# Endemism shapes viral ecology and evolution in globally distributed hydrothermal vent ecosystems

Marguerite V. Langwig[1,2], Faith Koester [1], Cody Martin [1,3], Zhichao Zhou [1], Samantha B. Joye[4], Anna-Louise Reysenbach [5] & Karthik Anantharaman [1,6,7] ✉

Viruses are ubiquitous in deep-sea hydrothermal vents, where they influence microbial communities and biogeochemistry. Yet, viral ecology and evolution remain understudied in these environments. Here, we identify 49,962 viruses from 52 globally distributed hydrothermal vent samples (10 plume, 40 deposit, and 2 diffuse flow metagenomes), and reconstruct 5708 viral metagenome-assembled genomes, the majority of which were bacteriophages. Hydrothermal viruses were largely endemic, however, some viruses were shared between geographically separated vents, predominantly between the Lau Basin and Brothers Volcano in the Pacific Ocean. Geographically distant viruses shared proteins related to core functions such as structural proteins, and rarely, proteins of auxiliary functions involved in processes such as fermentation and cobalamin biosynthesis. Common microbial hosts of viruses included members of Campylobacterota, Alpha-, and Gammaproteobacteria in deposits, and Gammaproteobacteria in plumes. Campylobacterota- and Gammaproteobacteria-infecting viruses reflected variations in hydrothermal chemistry and functional redundancy in their predicted microbial hosts, suggesting that hydrothermal geology is a driver of viral ecology and coevolution of viruses and hosts. Our results indicate that viral ecology and evolution in globally distributed hydrothermal vents is shaped by endemism and thus may have increased susceptibility to the negative impacts of deep-sea mining and anthropogenic change in ocean ecosystems.

An estimated $10^{30}$ viruses in the world's oceans are predicted to lyse and kill 20% of microbial biomass daily[1]. In marine systems, global sampling efforts have resulted in the recovery of viruses at a broad scale, enabling investigations of their ecology, evolution, and biogeography[2–4]. These studies revealed that viruses in the epipelagic ocean are passively transported via ocean currents[5], distinct viral groups exist across five oceanic ecological zones[4], and broad differences in the protein content of viruses occur across depth[6]. Though these studies provide valuable insights on viruses globally, they have mainly focused on comparative analyses of photic and aphotic viruses. Investigations of viruses in the deep ocean are lacking.

In deep-sea hydrothermal vents, viruses are an important source of predation on chemolithoautotrophic microorganisms at the base of the food web[7]. As a result of these infections, viruses have the potential

[1]Department of Bacteriology, University of Wisconsin-Madison, Madison, WI, USA. [2]Freshwater and Marine Sciences Program, University of Wisconsin-Madison, Madison, WI, USA. [3]Microbiology Doctoral Training Program, University of Wisconsin-Madison, Madison, WI, USA. [4]Department of Marine Sciences, University of Georgia, Athens, GA, USA. [5]Department of Biology, Portland State University, Portland, OR, USA. [6]Department of Integrative Biology, University of Wisconsin-Madison, Madison, WI, USA. [7]Department of Data Science and AI, Wadhwani School of Data Science and AI, Indian Institute of Technology Madras, Chennai, TN, India. ✉e-mail: karthik@bact.wisc.edu

to change microbial community composition and population sizes. Hydrothermal vent viruses are also capable of "reprogramming" host metabolism using auxiliary metabolic genes (AMGs). For example, vent viruses were the first viruses discovered to encode reverse dissimilatory sulfite reductase (rdsr), which can manipulate sulfur-based chemolithotrophy in the dark ocean[8]. To date, few studies have conducted comparative analyses of hydrothermal vent viruses globally, and those completed provide evidence that viruses are endemic to vent sites and habitat types at the genus level and that viruses infect ecologically important, abundant taxa such as Gammaproteobacteria and Campylobacterota[9]. Other studies have suggested vent viruses are predominantly lysogenic[10], have limited dispersal, and narrow host ranges[11].

Despite these advances, the factors controlling hydrothermal vent virus biogeography, ecology, and evolution remain poorly constrained. The biogeography of viruses is thought to be determined by a complex interplay between abiotic factors, virus traits (e.g., life cycle, virion size, burst size), and host traits (e.g., abundance, size, distribution)[12]. Abiotic factors such as vent chemistry have been shown to dictate microbial community composition, where differences in the chemical profiles of geographically close vent sites can result in distinct microbial communities[13,14]. Given their dependence on microbial hosts, viral biogeography is intimately linked to theories on microbial biogeography, however, these remain in their infancy[15,16]. Increasingly available hydrothermal vent metagenomic data presents opportunities to examine vent viral biogeography, ecology, and evolution at a global scale. This, coupled with advances in software for rapid, accurate comparison of microbial metagenome-assembled genomes (MAGs)[17] and the generation of viral MAGs (vMAGs)[18], promise to enable more accurate representations of environmental viruses and allow finer resolution comparisons of their community structure and ecology.

In this study, we catalog and describe viruses, largely from two types of hydrothermal vent environments, on a global scale: high-temperature hydrothermal deposits that host biofilms of thermophilic Bacteria and Archaea, and hydrothermal plumes hosting psychrophilic and mesophilic Bacteria and Archaea. These samples have previously been investigated for microbial diversity[13,14,19], but their viral communities are largely unexplored. Using 10 hydrothermal plumes, 40 vent deposits, and 2 diffuse flow samples collected from seven distinct hydrothermal systems in the Pacific and Atlantic Oceans (Guaymas Basin, Mid-Cayman Rise, Mid-Atlantic Ridge, Axial Seamount, Brothers Volcano, East Pacific Rise, and the Lau Basin), we leverage metagenomics and statistical analyses to reconstruct viral genomes and study viral communities through inter- and intra-vent comparative analyses. Our findings demonstrate that endemism is a key driver shaping the ecology and evolution of hydrothermal vent viruses through distinct viral and microbial traits.

## Results

We identified 63,826 viral scaffolds from 10 hydrothermal plumes, 40 hydrothermal vent deposits, and 2 diffuse flow metagenomes (Supplementary Data 1). Following viral identification, we conducted viral genome binning to reconstruct 5708 vMAGs. The vMAGs comprise 19,572 viral scaffolds (30.7% of scaffolds), leaving 44,254 unbinned viruses (69.3% of scaffolds). Thus, after viral binning, we recovered a total of 49,962 viruses from globally distributed hydrothermal vents (Fig. 1A). Of these, 1833 were characterized as medium-, high-quality, or complete (Supplementary Fig. 1), and 20,305 viruses encoded one or more viral hallmark genes (Supplementary Data 2). Most of the hydrothermal vent viruses were classified as lytic rather than lysogenic (47,571 lytic versus 2391 lysogenic), and this remains true when only examining viruses of medium-quality or better (1505 lytic versus 328 lysogenic), as well as only complete viruses (109 lytic versus 4 lysogenic). In addition, 32,442 vent viruses had a genome size range of

1–5 kb, while the remaining had genome sizes of 6–561 kb (Supplementary Fig. 2). Taxonomic predictions at the class level showed that most viruses are double-stranded DNA viruses within the realm *Duplodnaviria* (39,056), class *Caudoviricetes* (Supplementary Fig. 3). *Caudoviricetes* viruses were also the most abundant class of viruses in the dataset based on relative abundance (Fig. 1B and Supplementary Data 3). Most viruses were reconstructed from deposit samples from Brother's Volcano and the Lau Basin (35,094 viruses). These sites produced some of the largest assemblies of the datasets analyzed here (up to 1.2 Gb) and were the most intensively sampled compared to other sites (Supplementary Data 4).

### Viruses are rarely shared between geographically distant hydrothermal vents or the global ocean

To understand how hydrothermal vent viruses are related, we conducted similarity analyses of viruses across and within hydrothermal vents. We used clustering analysis based on average nucleotide identity (ANI), as well as read mapping of genomes. Clustering of ≥ 3 kb viral genomes identified 866 non-singleton clusters containing 1950 viruses, and most clusters (687/866) were composed of two viral genomes (Supplementary Data 5). Thus, most vent viruses were not included in ANI-based clusters, suggesting they have low relatedness at the nucleotide level. In addition, no clusters contained viruses from both hydrothermal plumes and hydrothermal deposits, indicating these habitats support distinct viral communities. Some viruses fell within the same nucleotide clusters, yet were reconstructed from geographically distant sites (Fig. 2A). Specifically, 65 clusters contained 152 viruses from geographically distinct vent deposits, with an ANI ranging from 70–99%. When examining the clusters, we found that the predicted viral genome sizes, completeness, hosts, and lifestyles were largely aligned. Most of the clusters (51/65) contained viruses from the Lau Basin and Brothers Volcano, which are both located in the South Pacific Ocean. Viral genomes in these clusters shared significant overlap even across geographically separated vents in different ocean basins, such as between the Mid-Atlantic Ridge in the Atlantic Ocean and Brothers Volcano in the Pacific Ocean, or between Axial Seamount and Guaymas Basin in the Pacific Ocean (Supplementary Data 5). Although viruses in the 65 geographically distinct clusters are predominantly predicted to be low-quality, we determined that the regions of overlap in these viruses are significant in length and/or annotation, and thus we find support for shared viral genomic regions between geographically separated vents (See Supplementary Note 1 and Supplementary Data 6 and 7).

Some shared viruses may be missed when examining viruses identified from metagenomes, because viral sequences may only be present in the reads and not the assemblies. To address this issue, we also used read mapping of all ≥ 3 kb viruses to identify those that were present in multiple samples. While nucleotide clustering indicated that no viruses were shared between hydrothermal vent plumes and deposits, read mapping-based detection identified 36 such viruses (Fig. 2B and Supplementary Data 8). Most of the read mapping analyses identified viruses shared between Lau Basin (26/36) deposits or plumes and the same or different sites in Lau Basin. These results suggest that some viruses may be shared between vent deposits and plumes.

To understand hydrothermal vent virus biogeography more broadly, we used nucleotide clustering to compare the viruses recovered here to the Global Ocean Virome[4] (GOV 2.0, see methods). This analysis revealed 36 clusters where hydrothermal vent viruses exhibited ≥ 70% ANI with those from the GOV 2.0 dataset (Supplementary Data 9). In some cases, these viruses were sampled in relatively close geographic proximity, such as a Guaymas Basin virus with 79% ANI to a virus from Tara Oceans Station 137, ~ 1575 km away. However, we also observed some instances where distant locations had similar viruses. For example, a complete genome of a Lau Basin (South Pacific Ocean)

virus had 81% ANI to a virus from Tara Oceans Station 68 in the South Atlantic Ocean. These viruses were also collected at distinct depths, and in some clusters, we observed viruses that were separated by over 2000 m depth. Thus, although we largely observed endemism in vent viruses, we observed rare cases where vent viruses were distantly related to viruses from different ocean basins and/or depths.

## Viruses are more similar within a hydrothermal field

In addition to understanding how viruses are related between geographically distant vents, we examined intra-vent viral relatedness, where 417 clusters contained 992 viruses from distinct sites within a vent field (Fig. 2A and Supplementary Data 5). Just over half of the clusters (272/417) contained viruses that were related between

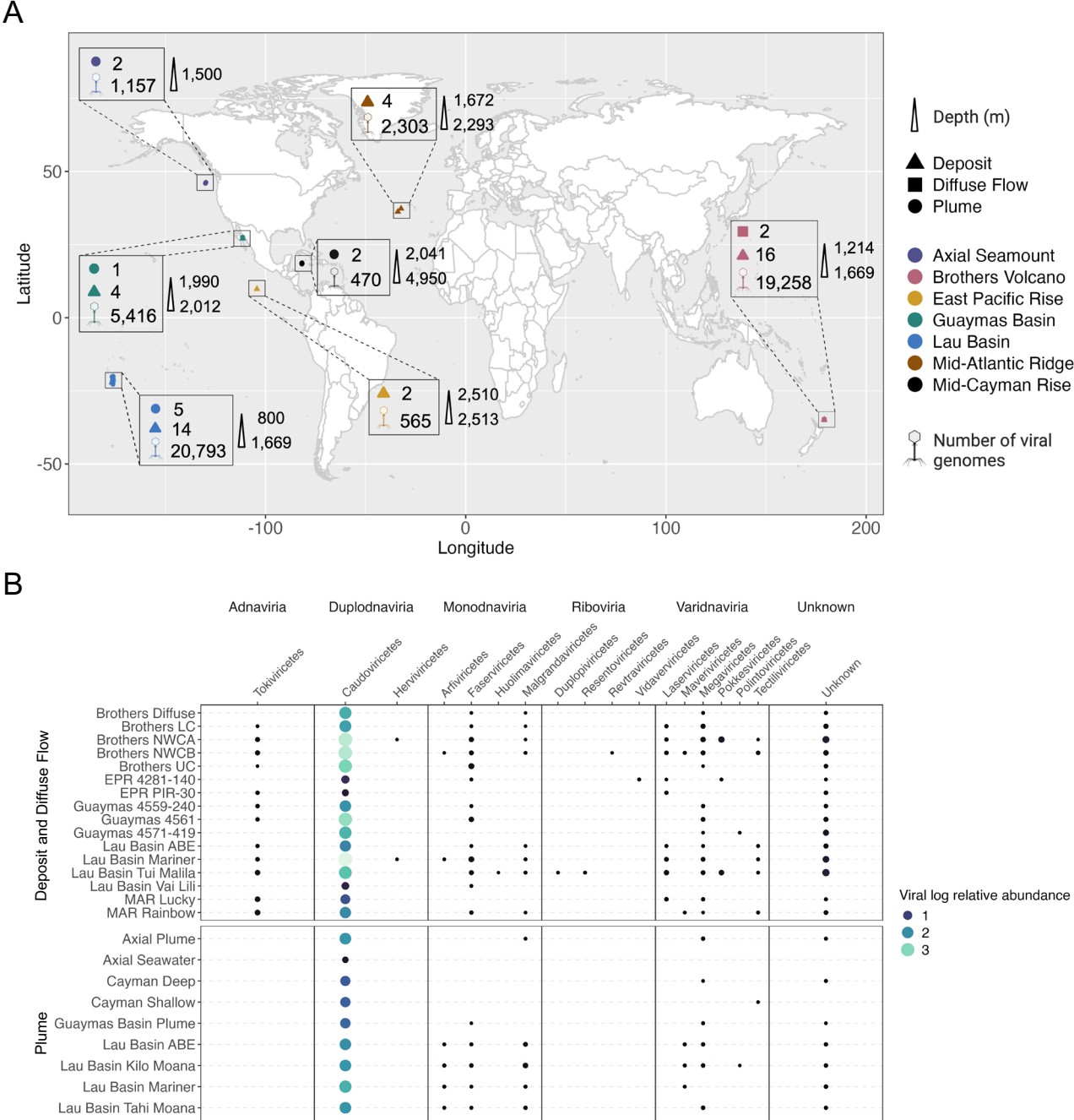

**Fig. 1 | Geographic distribution, abundance, and taxonomy of viruses identified in globally distributed hydrothermal vents. A** A world map showing the number of viruses identified from different hydrothermal vents. Circles represent metagenomic samples reconstructed from hydrothermal plumes, triangles are metagenomic samples reconstructed from hydrothermal vent deposits, and squares are diffuse flow. The numbers next to the shapes represent the number of samples for that vent field (52 total). Virus icons show the number of viruses identified at a vent site (49,962 total). Colors represent the seven distinct hydrothermal vent fields that are shown in the legend. The depth range of samples is shown to the right of the boxes and labeled with an elongated triangle. This figure was modified using BioRender.com (**B**). A bubble plot of log-transformed virus relative abundance, summed by viral class. Circles represent the log relative abundance, where larger relative abundance is represented by larger, teal circles, and smaller relative abundance is represented by smaller, dark blue circles. Virus class names are shown on the top *x*-axis, and the horizontal names above them show virus realms (determined using geNomad). Site names are shown on the left *y*-axis and the vertical names to the left of them show sample type.

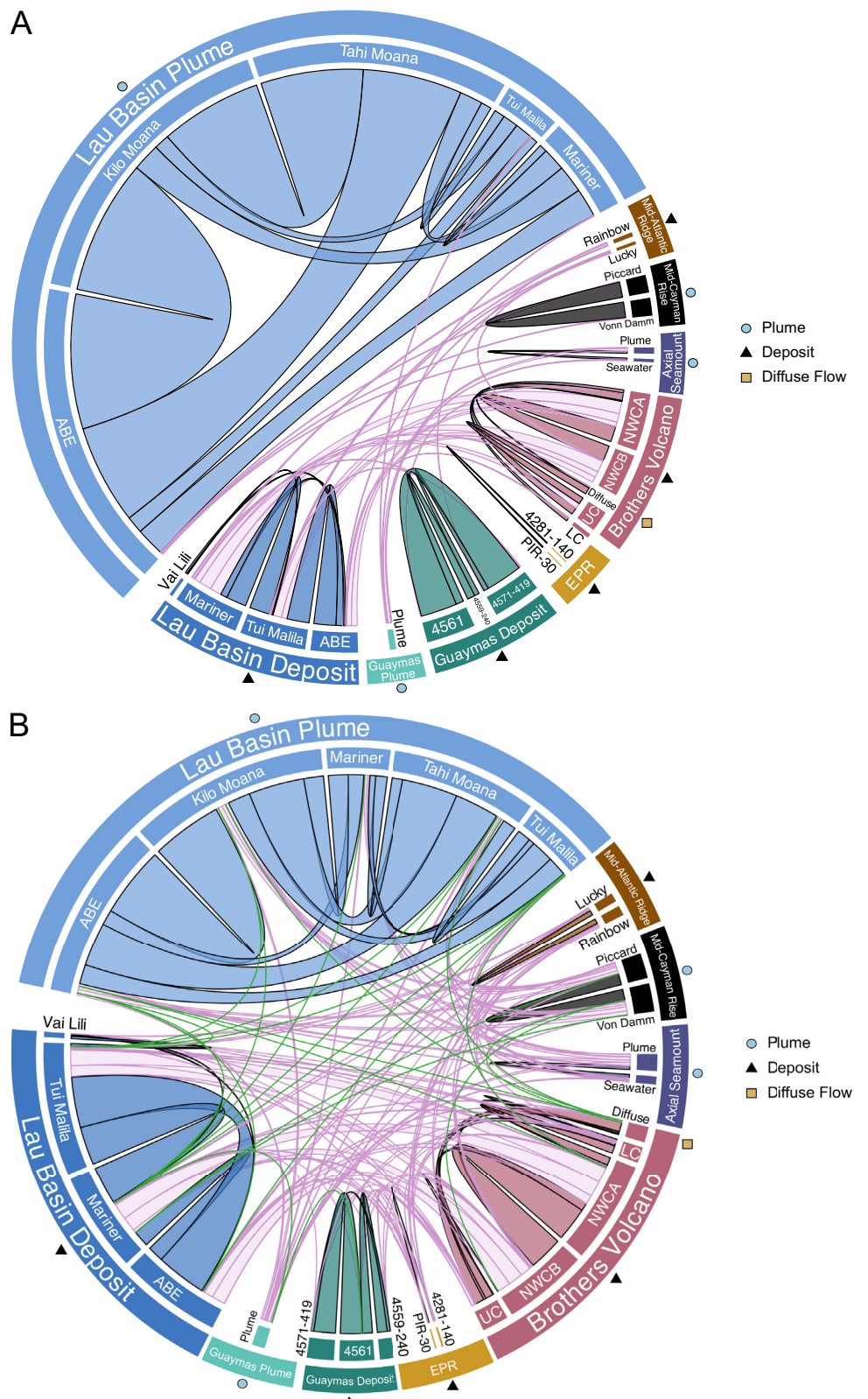

hydrothermal plumes in Lau Basin, including Lau Basin ABE, Kilo Moana, Mariner, Tahi Moana, and Tui Malila. These plumes are at most separated by 1–2° latitude (Supplementary Data 1). In the Lau Basin, 163/272 plume viral clusters had an average intra-cluster ANI ≥ 85%, and 90/272 had an average ANI ≥ 95%. This indicated that many of these Lau Basin plume viruses were related at the genus and species level, or for those with 100% identity and 100% completeness, were

identical. Almost all the viruses in Lau Basin plume clusters were predicted to be lytic and were represented by *Microviridae*, *Caudoviricetes*, *Inoviridae*, *Schitoviridae*, *Cressdnaviricota*, and *Demerecviridae*.

Lau Basin vent deposit samples had the next highest number of viral clusters shared between different vent fields (94/417 clusters, Supplementary Data 5). Of the 94 clusters, 16 contained medium-quality or better viruses, and nearly all of these were *Caudoviricetes*.

**Fig. 2 | Biogeography of hydrothermal vent viruses. A** Viral relatedness based on average nucleotide identity of ≥ 3 kb viruses and mcl clustering. Ribbons signify clusters that have viruses shared between hydrothermal vent sites (≥ 70% ANI). The width of the ribbon represents the number of clusters containing viruses from each site. Light pink ribbons are clusters of viruses from geographically distinct vent sites, while ribbons with the fill color of the site indicate clusters of intra-vent-related viruses (viruses from the same vent field but distinct vent locations). On the outer ring, circles show plume samples, triangles signify deposit samples and squares show the diffuse samples. **B** Viral relatedness based on read mapping between all reads and all viral genomes ≥ 3 kb length and ≥ 70% coverage. The width of the ribbon represents the number of times reads from one site mapped to a virus from another site. Light pink ribbons highlight reads that mapped between a vent and a virus from geographically distinct sites and green ribbons highlight reads that mapped between vent plumes and deposits. Ribbons with the fill color of the site indicate instances where reads from a vent site mapped to viruses from the same vent field (intra-vent read mapping).

Guaymas Basin and Brothers Volcano vent deposits, respectively, also contained many viral clusters shared between different vent fields. In Guaymas Basin, most clusters were shared between sites 4561 (380 and 384) and 4571-419 (38/47), which are geographically close but separated by a depth of 22 m. Similarly, in Brothers Volcano, many clusters were shared between Northwest Caldera Wall A (NWC-A) and Northwest Caldera Wall B and Upper Caldera Wall (NWC-B + UCW) (13/30), which are geographically close to each other (~1570 m distance), though they have distinct microbial communities[13]. Further, the Mid-Cayman Rise, Axial Seamount, and the East Pacific Rise had some intra-vent related viruses. To complement our clustering analyses, we also conducted read mapping-based detection, which indicated that viruses were related between intra-vent sites. Viruses from Lau Basin plumes were highly related (especially Kilo Moana and Abe, Tahi Moana and Kilo Moana). We also identified many viruses that were shared between Lau Basin deposits (Tui Malila, ABE, and Mariner), Brothers Volcano deposits (NWC-A and NWC-B), Guaymas Basin deposits (4571-419 and 4561), Mid-Atlantic Ridge deposits (Lucky and Rainbow), and Mid-Cayman Rise plumes (Von Damm and Piccard). Overall, read mapping-based detection reaffirmed viral relatedness between sites identified with ANI clustering but also identified numerous connections that were not observed through the ANI clustering analysis. This was especially true for overlap between viruses in hydrothermal plumes and deposits, viruses from the Mid-Atlantic Ridge, and viruses between more geographically distant vent fields like Kilo Moana and Cayman Shallow plumes since these patterns were uniquely observed with read mapping.

**Hydrothermal vents share viral protein families dominated by proteins of unknown function**

To understand how hydrothermal vent viruses are related at the protein level, we clustered all 595,416 vent virus proteins. This produced 74,940 clusters of two or more virus proteins. Of these, 152 clusters contained proteins shared between vent deposits and plumes (773 proteins), and 23,351 clusters have proteins shared between geographically separated vents (84,223 proteins or 14.2% of the protein dataset; Fig. 3A and Supplementary Data 10). Of the 84,259 total proteins shared across distant sites or sample types, 40,645 were annotated (48.2%), largely as hypothetical or uncharacterized functions, viral terminases, DNA/RNA polymerases, capsid, baseplate, tail, and portal proteins (Fig. 3B). These results were supported using a virus-specific protein database (PHROGs v4.0, see methods), where 49.8% of the annotations were characterized as unknown in function (Supplementary Data 11). Of the 152 clusters shared between vent plumes and deposits, 89 contained proteins with annotations, and the top annotation categories included helix-turn-helix domains, domains of unknown function, phage tail tube protein, hypothetical proteins, nucleotide kinase, and essential recombination function protein. The largest cluster of proteins from geographically separated vents had 88 proteins from deposit samples, including Brothers Volcano, the Lau Basin, and the Mid-Atlantic Ridge. This cluster contained a protein of unknown function that is known to often be encoded in phage genomes (PF09343 or IPR011740). Several of the next largest protein clusters contained proteins from the same sites (Brothers Volcano, Lau Basin, and Mid-Atlantic Ridge) that are of unknown function or AAA domain (PF13479). Many clusters with proteins from distant sites

were also functionally related to pyrimidine metabolism (e.g., dCTP deaminase, dTMP kinase, and dUTP diphosphatase), purine metabolism (phosphoribosylformylglycinamidine cyclo-ligase and phosphoribosylamine-glycine ligase (PurD), and nucleotide metabolism (ribonucleotide reductase). Several clusters contained phosphate starvation inducible proteins (PhoH), which have previously been identified as widespread in marine phages and proposed as a marker of marine phage diversity[20]. Some protein clusters that were highly similar between viruses from geographically distant vents contained proteins that do not have known core functions, including pyruvate formate-lyase activating enzyme (PflA) and cobaltochelatase subunits CobS and CobT (Supplementary Data 10). PflA is a key enzyme in fermentation and has been identified in a mimivirus, TetV-1[21]. CobS and CobT are involved in the synthesis of cobaltochelatase for cobalamin biosynthesis and have been identified previously in viruses[22]. Viruses have been hypothesized to use CobST to hijack host CobN, allowing for increased cobalamin production and synthesis of viral progeny.

**Viral biogeography is closely tied to the geographic distribution and abundance of their hosts**

To explore microbial drivers of viral biogeography, we predicted the microbial hosts and calculated the relative abundance of all viruses and their hosts (Supplementary Data 12, 13, and 14). Of the 49,962 viruses, 14% had a predicted host (7001 viruses, Supplementary Fig. 4). For these viruses, the infection range was largely narrow where, 6387 viruses were predicted to infect one host, and the remaining 614 viruses were predicted to infect >1 host. Most host predictions were for deposit viruses (84.7%), which are predicted to infect a greater diversity and larger number of microbial phyla compared to viruses in plumes (Supplementary Note 2). Among plume viruses, most predicted hosts were members of the phyla Pseudomonadota (formerly Proteobacteria, 44.6%) and Bacteroidota (17.2%), while in deposits, most were members of Campylobacterota (25%) and Pseudomonadota (21.2%, primarily Gamma- and Alphaproteobacteria). This aligned with relative abundance data, where the most abundant plume virus (Axial Plume, 1.3% relative abundance) infected a Proteobacteria in the class Gammaproteobacteria, genus *Thioglobus*, or SUP05. This sulfur-oxidizing bacterium is abundant in hydrothermal plumes globally[23,24]. In deposits and diffuse samples, the most abundant virus with a predicted host (Brothers Volcano site Diffuse, 1% relative abundance) was predicted to infect a member of Campylobacterota in the family Sulfurimonadaceae (genus *CAITKP01*). Bacteria in this family and genus are the most abundant among all the deposit samples, and isolates in this family from hydrothermal vents are known to be chemolithoautotrophic sulfur oxidizers[14,25].

**Hydrothermal geology and chemistry drives coevolution of viruses and hosts**

In both deposits and plumes, microbial hosts from the phylum Pseudomonadota were largely associated with the class Gammaproteobacteria. Previously, using the same metagenomes in this study, functional redundancy was observed between members of Gammaproteobacteria and Campylobacterota, where these taxa shifted as dominant community members depending on hydrothermal geology and chemistry. These microbial lineages had similar metabolic potential, and thus, their dominance at one hydrothermal vent site or

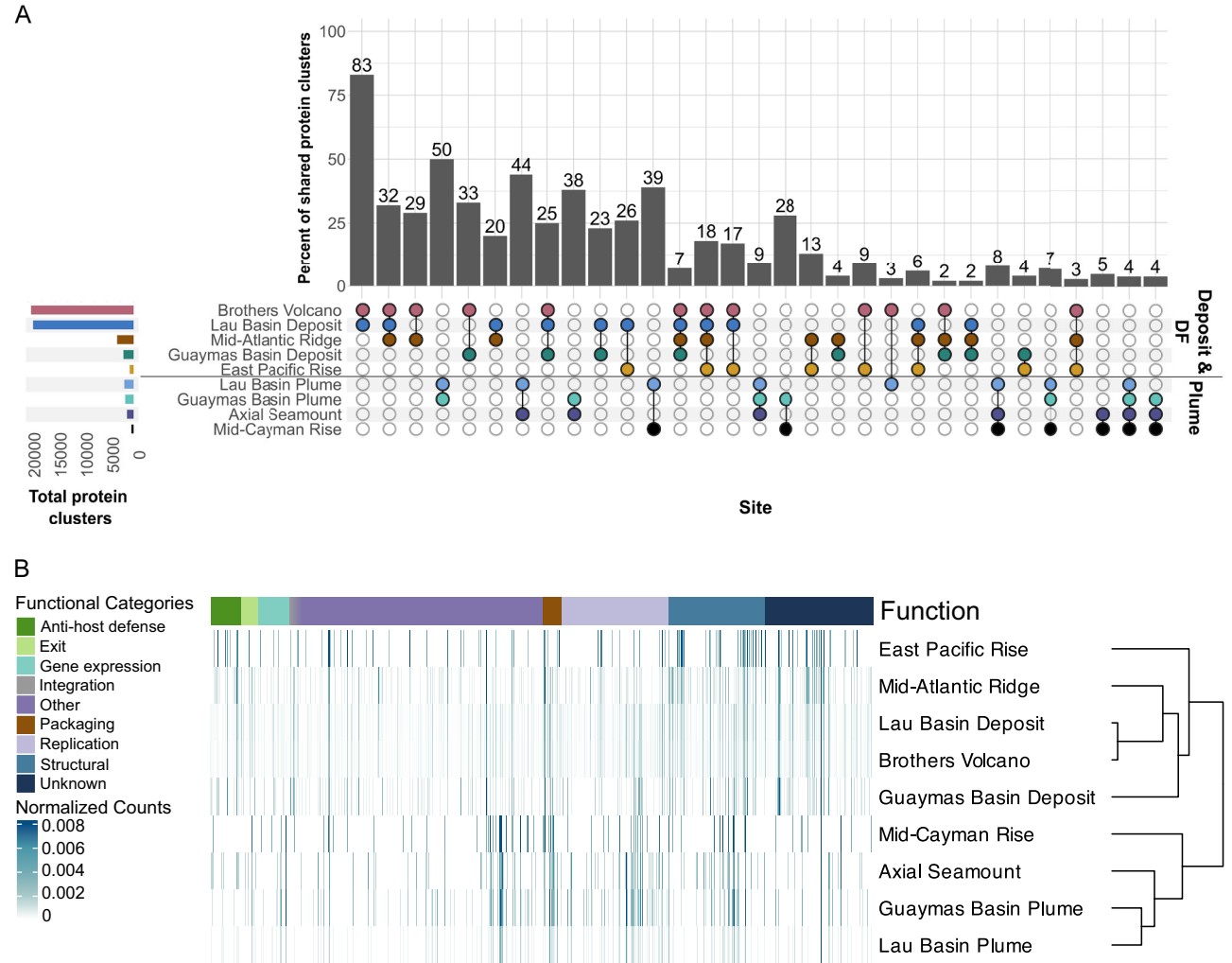

**Fig. 3 | Shared viral proteins between geographically distant hydrothermal vents. A** The bar plot shows the percent of shared protein clusters, and the bottom matrix shows the identity of the sites with shared protein clusters (filled, colored circles). The percent of shared protein clusters was calculated as the number of shared protein clusters divided by the smallest number of total protein clusters for a group, multiplied by 100. The leftmost bar plot shows the total number of protein clusters per site. The black line through the matrix separates deposit and diffuse flow (DF) samples from plume samples. Sites with fewer than fifteen shared protein clusters were removed. All clusters are reported in Supplementary Data 10. **B** A heatmap of the annotations of proteins shared between different hydrothermal vents or sample types. The annotations are the best hit of KEGG, VOG, and Pfam databases and were assigned functional categories using a custom Python script (see methods). The normalized counts were obtained by dividing the number of proteins per site by the total number of annotated, clustered proteins at that site. The dendrogram was produced by hierarchical clustering using the correlation distance "Spearman".

another was attributed to ecophysiological and growth differences, or distinct metabolic machinery for the same metabolic pathway[14]. Given this observation and our findings of the ubiquity of viral hosts from Gammaproteobacteria and Campylobacterota, we investigated patterns of relative abundance in the viruses that infect them. We found that Gammaproteobacteria- and Campylobacterota-infecting viruses reflected abundance patterns of their host (Fig. 4). For example, Campylobacterota-infecting viruses were abundant in vent deposits at Brothers Volcano site NWC-A, however, there was a shift to more abundant Gammaproteobacteria-infecting viruses at the geographically close site Brothers Volcano NWC-B. This was reflected in the abundance of the microbial host taxa (Fig. 4). Consistent with these qualitative patterns, compositional correlational analysis exhibited statistical congruence between Campylobacterota and the viruses that infect them, as well as Gammaproteobacteria and their viruses (see methods, Supplementary Data 15). Campylobacterota and Campylobacterota-infecting viruses exhibited relatively high proportionality values ($\rho_P = 0.90$)[26], and similarly, Gammaproteobacterota and Gammaproteobacterota-infecting viruses were correlated but

exhibited a weaker relationship ($\rho_P = 0.67$). From the clustering results, we also identified 55 viruses from geographically separated vents that had a predicted host. Of these, 25 were predicted to infect Campylobacterota or Gammaproteobacteria (Supplementary Data 5), further underscoring the importance of these microorganisms in driving viral coevolution in hydrothermal vents. In contrast to these taxa, phyla such as Aenigmatarchaeota, Micrarchaeota, an unknown bacterial phylum (EX4484-52), and Iainarchaeota were low abundance microbial community members (less than 1% relative abundance at all sites, Supplementary Data 14). In line with this, the viruses predicted to infect these microorganisms were few in number (29 viruses total), were not present in the nucleotide clusters (inter or intra vent), and were low in relative abundance.

### Hydrothermal viruses encode auxiliary metabolic genes associated with redox processes and detoxification
The presence of auxiliary metabolic genes, or AMGs, in viruses, may increase a virus' potential geographic range[12]. Viruses that encode AMGs have an increased ability to boost energy levels for viral progeny

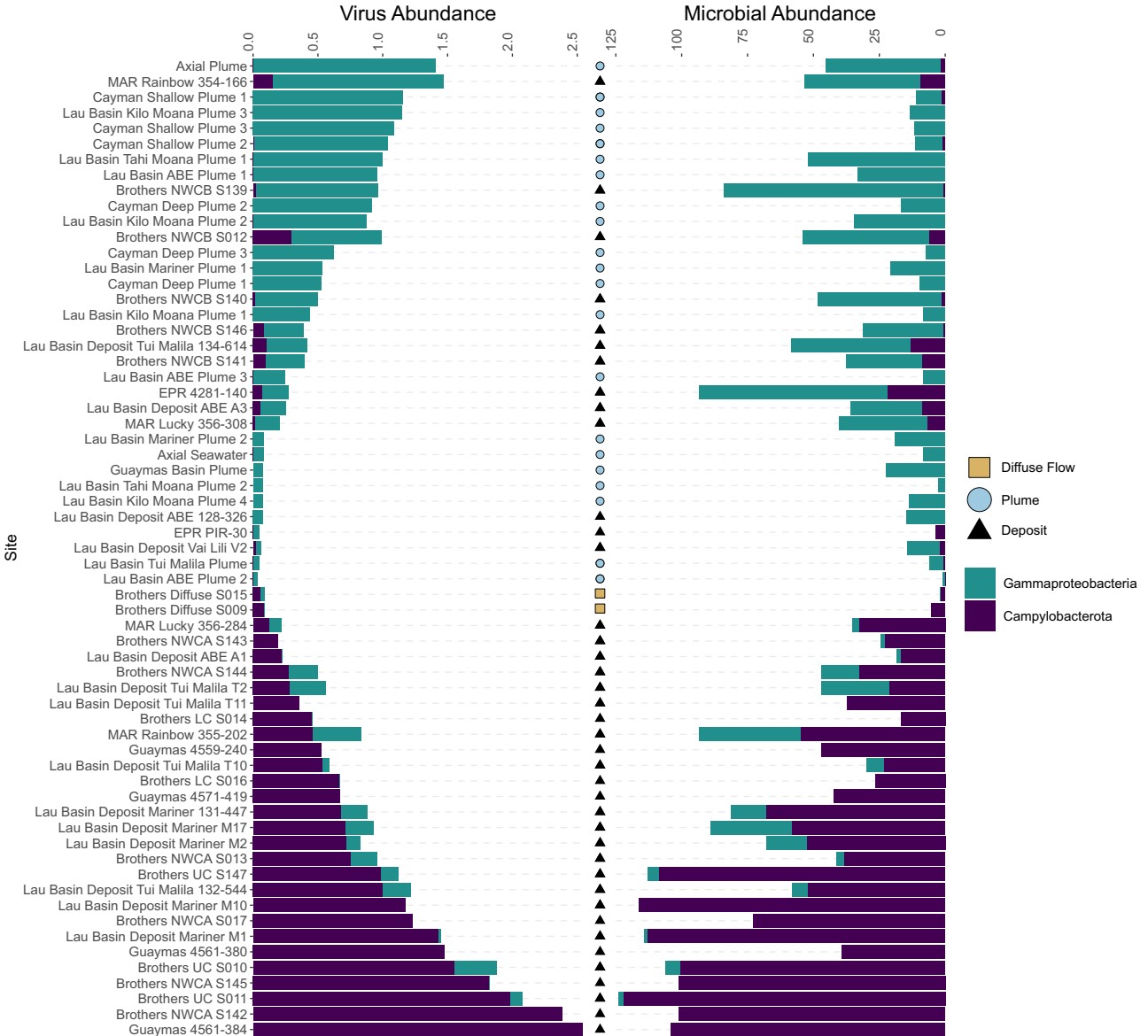

**Fig. 4 | Viral abundance of Gammaproteobacteria- and Campylobacterota-infecting viruses mimics functional redundancy of the hosts.** The relative abundance of viruses infecting Gammaproteobacteria and Campylobacterota is shown on the left, while the relative abundance of Gammaproteobacteria and Campylobacterota MAGs is shown on the right. Both abundances are the result of CoverM read mapping normalized by the number of reads in each sample. Sites are shown on the y-axis. The colors of the stacked bar plot show the type of host a virus infects (left plot) or the microbial taxa (right plot), and shapes indicate the sample type.

production or reduce the viral latent period and, thus, should be able to disperse more widely than viruses without AMGs. To investigate these dynamics, we searched for AMGs in all viruses using DRAM-v (distill mode, Supplementary Data 16). This allowed us to verify that AMGs were flanked by genes of viral or viral-like origin and were not present on the ends of genomic scaffolds. AMGs identified in the hydrothermal vents examined here were involved in sulfur metabolism, arsenic metabolism, nitrogen metabolism, and central carbon metabolism.

AMGs were rare in our dataset. According to DRAM-v, 2615 viruses encoded one or more AMGs, or ~ 5% of vent viruses. We identified a lytic *Caudoviricetes* virus reconstructed from Brothers Volcano that encoded adenylylsulfate reductase (AprB, K00395), which catalyzes the reduction of adenylyl sulfate to sulfite in the dissimilatory sulfate reduction pathway, or the reverse reaction in dissimilatory sulfur oxidation. A lysogenic *Caudoviricetes* virus from Lau Basin Mariner encoded an arsenate reductase (ArsC, K00537), which functions in arsenate

detoxification by reducing As(V) to an excretable form, arsenite or As(III)[27]. ArsC has been identified in soil viruses, where there is evidence that *arsC*-encoding viruses may contribute to metal resistance in their microbial host[28,29]. In addition to arsenate metabolism, we identified a virus encoding cytochrome bd ubiquinol oxidase subunit I and II (CydAB, K00425, and K00426), which acts as a terminal electron acceptor in the electron transport chain of microorganisms during respiration[30]. Although *cydAB* has not been described in other viruses, phage integration has been found to reprogram the regulation of anaerobic respiration in *Escherichia coli*, and thus CydAB may be another mechanism by which viruses manipulate host respiration[31]. Finally, a *Caudoviricetes* virus from the Lau Basin deposit was predicted to encode a nitric oxide reductase subunit B (NorB, K04561). NorB is the large subunit of nitric oxide reductase, which catalyzes the reduction of nitric oxide to nitrous oxide, the penultimate step of the denitrification pathway. NorB has previously been identified in viruses from an oxygen minimum zone in the Eastern Tropical South Pacific Ocean[32].

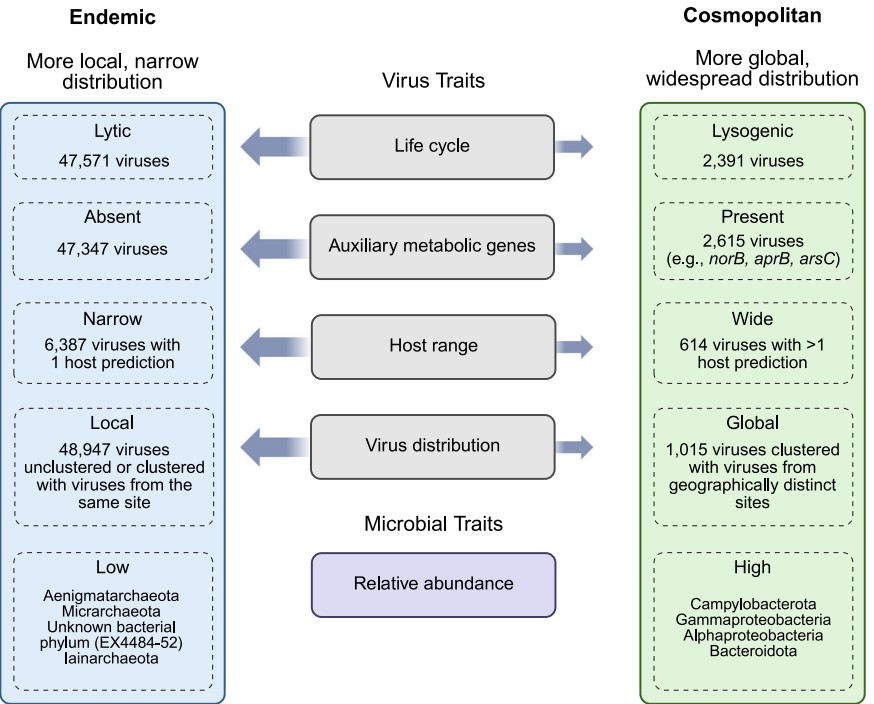

**Fig. 5 | Observed viral and microbial traits suggest endemism shapes the ecology and evolution of viruses in hydrothermal vents.** Conceptual diagram showing the viral and microbial traits (center, gray and purple rectangles) characterized in this study that contribute to viral biogeography in hydrothermal vents. Arrow width indicates the magnitude of support for local versus global viral distribution based on the findings in this study (large width signifies high support, small width indicates low support). The left blue rectangle shows traits associated with a more local, narrow distribution, and the right, green rectangle shows traits associated with a global, widespread distribution. This figure is adapted from Chow and Suttle (2015)[12] and was created using BioRender.com.

## Discussion

Hydrothermal vent viruses are known to be a key driver shaping microbial communities, yet they have remained understudied in these ecosystems. By analyzing viruses and microbes recovered from 52 globally distributed hydrothermal vent metagenomes, we show that endemism shapes viral ecology and evolution in deep-sea hydrothermal vents (Fig. 5). Few prior studies have investigated hydrothermal vent viruses at a global scale[9], and this is the first comparison of viral communities between hydrothermal vent chimney deposits and plumes. Furthermore, we incorporate viral binning to resolve vMAGs and thus provide a more accurate picture of viral diversity.

Most viruses identified in this study were characterized as lytic, a viral trait that has been suggested to limit a virus's distribution compared to lysogeny, where a virus can be dispersed within a host[12]. This finding is corroborated by the medium-, high-quality, and complete viruses that we recovered here, for which we also observe more lytic viruses than lysogenic. The lack of AMGs identified in this study may also limit viral dispersal, as viruses without AMGs would not be as well-equipped to boost energy levels for viral progeny production or reduce the viral latent period[33]. Similarly, the microbial host range was narrow for most viruses with a predicted host, which is thought to be limiting in dispersal compared to a wide host range (viruses able to infect >1 genera). Among microbial hosts, Campylobacterota and Gammaproteobacteria MAGs were widespread and abundant, suggesting the viruses infecting these hosts have a greater potential to disperse. Indeed, we found support for this in the relative abundance of viruses infecting these taxa, as well as their presence in nucleotide clusters from geographically separated vents. Conversely, viruses infecting low-abundance microorganisms were observed to be more rare and had limited dispersal. Thus, most hydrothermal vent viruses have traits that have been previously hypothesized to promote a narrow, local distribution.

The viral characteristics observed in this study provide a more holistic view of the hydrothermal vents analyzed here, which have previously been characterized for their microbial diversity[13,14,19]. In the hydrothermal plumes, prior studies showed that sulfur compounds dominated as an energy source, and plumes consisted of 14 core microbial genera, six of which were within the class Gammaproteobacteria. In deposits, many bacterial and archaeal genera were identified as endemic, and Gammaproteobacteria and Campylobacterota were shown to exhibit functional redundancy associated with energy metabolism, including sulfur, nitrogen, hydrogen, and oxygen metabolism. This was hypothesized to arise from differences in the chemical profiles of different vents, which then selected for eco-physiological and growth differences between taxa[13,14]. These observations were reflected in the viruses recovered here, where Gammaproteobacteria-infecting viruses were abundant in hydrothermal plumes typically associated with lower concentrations of hydrothermal compounds such as hydrogen sulfide, sulfur, and hydrogen, while Camplyobacterota-infecting viruses were more abundant in deposits associated with higher concentrations of reduced hydrothermally-derived compounds. The viral abundance patterns thus reflected the functional redundancy of Campylobacterota and Gammaproteobacteria in hydrothermal systems. These results underscore the influence of geological context in driving the evolution of microorganisms, and the resulting coevolution of viruses with their microbial hosts. This coevolution likely contributes to high host specificity and the high levels of endemism we observed among viral populations.

The unique viral communities we observed in hydrothermal plumes versus deposits is consistent with previous microbial studies[34], as well as viral studies that have shown differentiation of viruses between sediments and plumes[9]. While microbial communities in deposits have been found to correlate with geochemistry[13,35,36], there is evidence that plume microbial communities do not[34]. Instead,

microbial communities in plumes of the Lau Basin were shown to be similar, despite differences in their geography, depth, and geochemistry. This aligns with our findings, where the most intra-vent virus similarity was identified in Lau Basin plumes, despite the smaller number of viruses recovered here compared to Brothers Volcano and Lau Basin deposits. Lau Basin plume connectivity has been hypothesized to be promoted by characteristics such as weak stratification and diapycnal mixing over rough topography[34]. Greater sampling resolution is needed within hydrothermal vent fields to dissect the role of local geography in promoting connectivity through hydrothermal plumes. These studies should also be conducted temporally, as hydrothermal systems are dynamic and are known to change drastically over short time periods based on seismicity[37]. Investigations of the same site over time will better elucidate how changes in geology drive the coevolution of viruses and microorganisms.

Functionally, viral proteins shared between different vents were predicted to be involved in genome replication, viral structural components, viral infection, lysogeny, and often, are of hypothetical or unknown function. Similar to previous work, our study was hampered by our inability to annotate a large number of viral proteins[38], where less than half of the proteins shared between different vents were annotated. Poor protein annotation rates inhibited our understanding of the mechanistic underpinnings of connectivity, and thus the unannotated proteins identified represent interesting targets for future work to better understand the core proteome of hydrothermal vent viral communities. Of those annotated, few of the proteins shared between geographically separated vents have auxiliary functions, such as those related to microbial dissimilatory metabolism, and these genes were not common in the dataset overall. Thus, although viral genes related to microbial energy metabolism pathways, such as sulfur oxidation, are known to occur in hydrothermal vents[8], these genes were rare in these ecosystems.

While our findings align with key insights from prior microbial and viral studies, it is also important to acknowledge several limitations in this work, which offer a foundation for future research into deep-sea viruses. First, our study is focused on prokaryote-infecting viruses from metagenomes, and thus largely leaves out eukaryote-infecting and RNA viruses. Currently, there are a lack of tools capable of predicting unicellular eukaryotic hosts and the hosts of giant viruses[39]. The recovery of transcriptomes from hydrothermal vents remains a challenge, though several recent studies have begun to characterize the diversity of RNA viruses in the epi- and mesopelagic ocean[3,40]. Second, predicting the lifestyle of viruses remains a challenge in the field of viromics, where lysogenic virus predictions are often dependent on the presence of proteins associated with lysogeny, such as integrases. These analyses can be dependent on genome completeness, though our findings of predominantly lytic viruses remain true when examining predicted medium-quality and above viruses, as well as only complete viral genomes. Similarly, we cannot rule out the possibility that low genome completeness plays a role in the lack of AMGs we identified. A recent study supports a low to moderate number of AMGs in marine prokaryotic viruses globally, where an estimated ~19–38% of ocean virus populations have at least one AMG[41], though this dataset did not include viruses below 1000 m depth. It is also possible that some viral genes with auxiliary functions were overlooked in our analyses because current tools largely rely on our knowledge of gene function from the prokaryotic domain. Most AMGs are acquired from their hosts, and over time, resulting proteins can have changes such as structural alteration, changes in their functional role, and the production of multiple proteins through overlapping genes[42,43]. Such phenomena can obfuscate the role of viral proteins, making AMGs difficult to identify and the function of putative AMGs difficult to ascertain. These aspects of viral ecology present exciting research opportunities for deep-sea viruses that warrant further exploration.

The high levels of endemism identified in hydrothermal vent viral communities in this study and prior work[9,11], combined with endemism identified in vent microorganisms and animals[44], suggests these ecosystems could be especially negatively impacted by future disturbance[45]. In the face of deep-sea mining and anthropogenic climate change, microbial diversity, biomass, and metabolic rates may be severely negatively impacted[46], and this, in turn, will be detrimental to viral communities. Specifically, mining activities will disrupt natural fluid flow from hydrothermal systems, potentially reducing the availability of chemical energy for primary-producing microbial chemolithoautotrophs, heterotrophs, and the larger food web. This will affect the viruses that infect these organisms. In addition, the large number of unknown viral proteins highlights the vast biological potential we stand to lose as a result. In the future, additional studies are needed to probe the functions of unknown viral proteins, investigate biogeographic patterns on a temporal scale, and obtain a better sampling resolution of hydrothermal vents. This will provide a better understanding of how deep-sea mining and other anthropogenic influences will impact hydrothermal vent communities, and how we can mitigate these disturbances.

## Methods

### Sample collection

This research complies with all relevant ethical regulations. All necessary permits and approvals for sample collection were obtained from the appropriate authorities, and all procedures were conducted in accordance with established ethical guidelines. Hydrothermal plume samples were collected from the corresponding cruises (Supplementary Data 1): R/V *New Horizon* in Guaymas Basin, Gulf of California (July 2004)[47–49], R/V *Atlantis* in Mid-Cayman Rise, Caribbean Sea (Jan 2012)[48], R/V *Thomas G. Thompson* in the Eastern Lau Spreading Center (ELSC) western Pacific Ocean (May-July 2009)[8,48], and R/V *Thomas G. Thompson* in Axial Seamount, Juan de Fuca Ridge, northeastern Pacific Ocean (Aug 2015)[50].

Guaymas Basin plume samples were collected by "tow-yo" casts using a CTD rosette in 10 L Niskin bottles[49]. This water was then filtered onto 142 mm 0.2 μm polycarbonate filters by $N_2$ gas pressure filtration and preserved in RNAlater[51]. Mid-Cayman plume samples were collected using a Suspended Particle Rosette Sampler (SUPR) by filtering 10–60 L plume water onto 142 mm 0.2 μm SUPOR membranes[48]. These samples were then preserved in RNAlater in situ. In Lau Basin, SUPR-collected samples were filtered onto 0.2 and 0.8 μm pore size SUPOR polyethersulfone membranes in situ and preserved in RNAlater-flooded vials[34]. In Axial Seamount, plume samples were collected by a Seabird SBE911 CTD and 10 L Niskin bottles[50]. Samples of 3 L were then transferred into cubitainers and filtered through 0.22 μm Sterivex filters.

Hydrothermal deposit samples were collected from the corresponding cruises: R/V *Thomas G. Thompson* in Brothers Volcano, western Pacific Ocean (March 2018)[13], R/V *Roger Revelle* (April and May 2015)[13] and R/V *Melville* (April 2005)[14] in the ELSC, western Pacific Ocean, R/V *Atlantis* in Guaymas Basin, Gulf of California (Nov and Dec 2009)[14,52], R/V *Atlantis* in the Mid-Atlantic Ridge, Atlantic Ocean (July 2008)[14], and R/V *Roger Revelle* in the East Pacific Rise, Pacific Ocean (March 2004 and December 2006)[14]. Once on the ship, deposit samples were subsampled with the outer few millimeters (up to approximately 5 mm) kept separate from the bulk sample. These exterior samples were homogenized and stored at −80 °C for subsequent DNA extraction[33].

### DNA extraction and sequencing

For Guaymas Basin, Mid-Cayman Rise, and Lau Basin plume samples, DNA was extracted from ¼ filters using chemical and physical lysis methods as described in Dick and Tebo (2010)[47] and Li and Dick (2015)[48], and sequenced with Illumina HiSeq2000 at the University of

Michigan DNA Sequencing Core. Axial Seamount plume samples were extracted using a phenol-chloroform extraction, and metagenomic libraries were constructed using the Ovation Ultralow Library DR multiplex system[50]. Sequencing was completed using a NextSeq 500 at the W.M. Keck sequencing facility, Marine Biological Laboratory, in Woods Hole, MA.

For Brothers Volcano and ELSC (2015) deposit samples, DNA was extracted from homogenized deposits using the DNeasy PowerSoil kit (Qiagen), and metagenomic libraries were constructed using Nextera DNA Library Prep kits (Illumina), as described in Reysenbach et al., (2020)[13]. Sequencing was completed at the Oregon State University Center for Genome Research and Computing on an Illumina HiSeq 3000. For ELSC (2005), MAR, EPR, and Guaymas Basin, DNA was extracted using the Ultra Clean Soil DNA Isolation Kit (MoBio Laboratories, Carlsbad, CA, USA)[36]. Metagenomic libraries were prepared and sequenced at the Department of Energy, Joint Genome Institute (JGI)[14].

## Metagenomic assembly and microbial binning

Hydrothermal plume assemblies and microbial MAGs were generated as described in Zhou et al., 2023[19]. Briefly, metagenomic assemblies were constructed from QC-processed reads with MEGAHIT v1.1.2[53] using the following parameters: --k-min 45 --k-max 95 --k-step 10. Plume assemblies from Mid-Cayman Rise, Lau Basin Abe, Mariner, and Tahi Moana represent combined plume and background seawater, meaning plume reads were co-assembled with background seawater reads. Microbial MAGs were generated using MetaBAT v0.32.4[54] using 12 combinations of parameters, followed by DAS Tool v1.0[55] to generate consensus MAGs. Following MAG refinement and contaminant removal, only MAGs with >50% completeness and <10% contamination were retained, as determined by CheckM v1.0.7[56].

Hydrothermal deposit assemblies and microbial MAGs were generated as described in Zhou et al., 2022[14]. Briefly, reads from Brothers Volcano and ELSC (2015) were quality-filtered using FastQC v0.11.8[57] and de novo assembled using metaSPAdes v3.12.0[58] with the parameters: -k 21,33,55,77,99,127 -m 400 –meta. Reads from ELSC (2005), MAR, EPR, and Guaymas Basin were assembled by the Department of Energy, Joint Genome Institute (JGI) using metaSPAdes v3.11.1[58] with the following parameters: -k 33,55,77,99,127 –only-assembler –meta. MetaWRAP v1.2.2[59] was used to generate microbial MAGs with parameters –metabat2 –metabat1 –maxbin2. DAS Tool v1.0[55] was then used to generate consensus MAGs.

## Virus identification and binning

VIBRANT v1.2.1[60] was run with default parameters to identify viruses from the genomic assemblies of the 52 hydrothermal vent samples, resulting in 63,826 viral scaffolds. Viral scaffolds were binned using vRhyme v1.1.0 on each of the 52 hydrothermal vent samples with default parameters and bam files[18]. The sorted bam files used in binning were generated by mapping the fastq reads for a particular sample to the genomic assembly reconstructed for the same sample. Specifically, a custom Python script was used to run BWA-MEM v0.7.17[61] to map reads to assemblies and samtools v1.7[62] to convert sam files to bam format and then obtain sorted bam files. In total, we reconstructed 49,962 viral genomes. vMAGs were screened for high protein redundancy and binning of lytic and lysogenic viruses, where those with ≥ 2 redundant proteins and/or ≥ 2 lysogenic scaffolds were broken back into individual scaffolds and retained in the dataset as unbinned viruses. Finally, vMAGs >10 scaffolds were retained in the dataset as unbinned viruses.

For finalized vMAGs, the vRhyme script link_bin_sequences.py was used with default parameters to generate one scaffold vMAGs, where each scaffold is linked by 1,500 Ns. This is required by some downstream tools that expect viral genomes to be on one scaffold (e.g., CheckV, iPHoP) as described below. Viral genome size was

determined using SeqKit v2.6.1 on unbinned viral scaffolds and binned viruses without N-links[63]. To visualize the number of viruses reconstructed from each site, Fig. 1 was generated in R v4.4.0[64] using a custom R script (see SitesMap.R at https://github.com/mlangwig/HydrothermalVent_Viruses)[65]. Viruses were designated as lytic or lysogenic based on VIBRANT, which uses the presence/absence of integrase or the excision of a viral region from metagenomic scaffolds to determine whether a virus is lytic or lysogenic. For vMAGs that had lytic and lysogenic scaffolds binned into one genome, the genome was designated as lysogenic (VentVirus_Analysis2.R).

## Virus taxonomy, marker genes, and quality

Virus taxonomy was determined using geNomad v1.5.1[66], which utilizes taxonomically informed marker genes to determine the most specific taxon supported by most of the viral genes in the genome. Taxa are defined according to the International Committee on the Taxonomy of Viruses (ICTV)[67]. The end_to_end pipeline was run with default parameters. This also allowed us to determine the number of viral hallmarks encoded by each viral genome. Virus genome quality was determined using the CheckV v0.8.1 end_to_end pipeline with default parameters to obtain predictions for low-, medium-, and high-quality as well as complete genomes[68]. For both geNomad and CheckV, vMAGs were input with scaffolds concatenated by 1500 Ns.

## Virus nucleotide clustering

The average nucleotide identity (ANI) of ≥ 3 kb viruses was calculated with skani v0.2.0[17] and clustered using the Markov Clustering Algorithm (mcl, release 14-137)[69]. Vskani v0.0.1 (available at https://github.com/cody-mar10/skani-vMAG) was used to run skani and mcl, treating vMAGs as genomes and unbinned viruses as single scaffolds, with the following parameters: vskani skani -c unbinned_PlumeVent_viruses.fna -d fna_vMAGs -x.fasta -m 200 -cm 30 -s 70 -f 50 -ma .7. Option -m signifies the number of marker k-mers used per bases and was lowered to 200 from the default 1000 due to the smaller genome size of viruses compared to microorganisms. The compression factor (parameter -cm, equivalent to -c or --slow in skani) was lowered from the default of 125 to 30 to provide more accurate estimates of aligned fractions (AF) for distantly related viral genomes. The screen parameter (-s) removed pairs with less than 70% identity, while the minimum aligned fraction parameter (-f) kept ANI values where one genome had an aligned fraction greater than or equal to 50%. Finally, -ma signifies minimum ANI, which was lowered to 70% from the default 95% to capture a broader range of viral relatedness (the family and genus level). Mcl clustering was completed using default parameters.

The resulting table of skani-produced ANI and AF values was modified using a bash script, where ANI was normalized by the lowest AF (ANI*AF/100²), filtered for ≥70% ANI (now corrected for AF), and formatted as input for the mcxload function of mcl. Mcl clustering was then run with default parameters. The resulting cluster file was input into R to calculate the average ANI per cluster, map metadata to the clusters, and determine which clusters contained viruses from geographically distinct sites (ANI_clust.R). The same nucleotide clustering methodology and R script were used to compare the viruses recovered in this study to the Global Ocean Virome 2.0 dataset[4], downloaded from iVirus on CyVerse (https://datacommons.cyverse.org/browse/iplant/home/shared/iVirus/GOV2.0).

To identify the regions of overlap between low-quality viruses, BLASTN v2.14.1[70] was run per cluster on all viruses within a cluster. First, a blast database was made for each cluster with the following command: makeblastdb -in -out -dbtype nucl. Then, blastn was run on each cluster with the following command: blastn -query -db -out -outfmt "6 qseqid sseqid evalue bitscore length pident qstart qend sstart send" -max_target_seqs 2 -max_hsps 1. The aligned coordinates were obtained from these output files. Finally, bedtools v2.31.0[71] was used to extract these coordinates from the amino acid-format virus

genome files with the following command: bedtools intersect -a bed_GeoDistinct_VirusClusts.tsv -b bed_PlumeVent_viruses.tsv -wa -wb > result.txt. The -a bed file contains the coordinates from the blastn results, and the -b bed file contains the coordinates of all the ORFs in all the viral genomes. This allowed us to obtain the gene annotations of the regions of the viral genomes that had identical nucleotides.

## Virus protein clustering

Mmseqs2 (v15.6f452)[72] was used to cluster viral proteins. First, the createdb option was used to create an mmseqs database of the 595,416 hydrothermal vent virus proteins. Next, the cluster option was used with the following parameters: -cov-mode 0 --min-seq-id 0.75. These options signify that the alignment covers at least 80% of the query and of the target and that the minimum sequence identity is 75%. The default clustering algorithm, the greedy set cover algorithm, was used. This algorithm iteratively selects the node with the most connections and all its connected nodes to form a cluster and repeats this process until all nodes are within a cluster. Next, the createtsv option was used to generate a tsv file of the cluster output. This file was parsed and analyzed in R (Protein_clustering2.R). The clusters were visualized in R with an UpSet plot using the ComplexUpset package (v1.3.3)[73].

## Protein annotations

Viral proteins were annotated using VIBRANT[60], which employs hmmsearch to annotate viral proteins with KEGG, VOG, and Pfam HMM databases. To determine the best-supported VIBRANT hits from the three databases, the annotation with the highest bit score was chosen, followed by the lowest e-value, and finally, the highest viral score. We also used the PHROGs database (v4.0)[74] to annotate viral proteins using default parameters. The output was filtered for hits with ≥ 80% coverage and ≥ 75% sequence identity, and these results were parsed for the best-hit protein using the smallest e-value and the largest bit score. Potential viral AMGs were identified using DRAM-v v1.4.6[75]. To run DRAM-v, VIBRANT-identified viruses were input into Virsorter2 v2.2.4[76] to obtain the input file needed for the DRAM-v software. Because Virsorter2 was used for downstream analyses and not viral discovery, we used the following parameters: virsorter run --keep-original-seq --prep-for-dramv --include-groups dsDNAphage,N-CLDV,RNA,ssDNA,lavidaviridae --provirus-off --viral-gene-enrich-off --min-score 0.0. DRAM-v annotate was run with the default parameters. DRAM-v distill was run with default parameters to obtain annotations that are supported as AMGs. Briefly, this mode produces an AMG summary file where genes are shown that have an auxiliary score of 1-3. This signifies that the genes with putative AMGs are flanked by two viral hallmark genes (score of 1), flanked by one viral hallmark gene and one viral-like gene (score of 2), or flanked by two viral-like genes (score of 3).

Protein annotations that were within clusters from geographically distant vents or sample types were visualized in a heatmap (Fig. 3B) using the ComplexHeatmap package[77] (v2.20.0) in R (Protein_clustering2.R). Annotations were summed by site, and these counts were normalized by dividing the number of annotations by the total number of proteins annotated at a site. Functional categories were assigned to the annotations from three separate databases using a Python script available at: https://github.com/cody-mar10/protein_set_transformer/blob/main/manuscript/protein_embeddings/annotations/relabel_VOG.ipynb.

## Read mapping

All ≥ 3 kb viral genomes were mapped to all 163 paired-end fastq reads using CoverM v0.6.1[78] (https://github.com/wwood/CoverM) with the options --methods count relative_abundance --min-covered-fraction 0. For read mapping used to determine connectivity between vents, the output was filtered in R to only retain viruses where reads mapped to ≥ 70% of the viral genome (Coverm_circos.R). To obtain normalized relative abundance, the number of reads mapped to a virus from each sample was divided by the number of reads in that respective sample. This methodology was repeated with the microbial MAGs to obtain their normalized relative abundances.

## Host prediction

iPHoP v1.3.3 was used to identify virus-host links between hydrothermal vent viruses and a custom database of 3872 MAGs reconstructed from the same sites[14,19,79]. To predict potential bacterial and archaeal hosts, the iPHoP tool leverages a combination of approaches, such as blastn matches to host genomes and CRISPR spacer databases, as well as k-mer composition similarity. Before building the custom database, BLASTN[70] was used to search all viruses against all microbial MAGs with the following command: nohup blastn -query -db -out -outfmt "6 qseqid length qlen slen pident bitscore stitle". Hits with 100% identity and 100% coverage were considered viral contamination (generated by misbinning) and were removed from microbial MAGs to prevent false positives based on blastn searches to the host genomes. Following this step, the custom MAG database was created using the add_to_db option, and iPHoP was run using the following parameters: iphop predict --db_dir --no_qc. The input file included unbinned viral scaffolds and vMAGs concatenated into one scaffold using 1500 Ns to enable one prediction per vMAG.

Following host prediction, viruses that infect Campylobacterota and Gammaproteobacteria were subset from the CoverM read mapping data and used to visualize their relative abundance patterns (shown in Fig. 4). To determine whether the observed qualitative relationship between their relative abundances was quantitatively supported, we calculated proportionality using the propr package in R (v5.1.4, AbunCorr.R)[26]. The input data consisted of the sum of the number of reads mapped to each group (Campylobacterota, Gammaproteobacteria, Campylobacterota-infecting viruses, and Gammaproteobacteria-infecting viruses). This input was used to calculate the association matrix using the proportionality metric "rho" ($\rho_p$) and the centered log-ratio (clr) transformation.

## Reporting summary

Further information on research design is available in the Nature Portfolio Reporting Summary linked to this article.

## Data availability

The viral genomes recovered in this study are available at https://figshare.com/articles/dataset/Hydrothermal_Vent_Viruses/25968037. The predicted medium-, high-quality, and complete viruses with geNomad-assigned taxonomy are available through NCBI BioProject ID PRJNA1183900. The microbial metagenome-assembled genomes are publicly available through NCBI BioProject IDs PRJNA488180 and PRJNA821212. All the raw reads are publicly available with the SRA and ENA ID numbers listed in Supplementary Data 1.

## Code availability

The scripts used in this work are available at https://github.com/mlangwig/HydrothermalVent_Viruses. The DOI for this GitHub repository is: https://doi.org/10.5281/zenodo.15028718.

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

## Acknowledgements

This research was supported by the National Science Foundation under grant numbers DBI-2047598 (to K.A.), OCE-2049478 (to S.B.J. and K.A.), and OCE-0728391, OCE-0937404, OCE-1558795 (to A.L.R.). C.M. was funded by a National Science Foundation Graduate Research Fellowship. We thank the crew of the R/V *Roger Revelle*, R/V *Atlantis*, R/V *Thomas G. Thompson*, HOV *Alvin*, and the ROV *Jason* for assistance in collecting these samples. Thank you to Spencer R. Keyser for help data wrangling in R.

## Author contributions

M.V.L., A.L.R., and K.A. conceptualized the project. K.A. supervised the project. K.A., S.B.J., and A.L.R. obtained and sequenced the hydrothermal samples. Z.Z. performed metagenomic assembly and binning. M.V.L. identified viruses from the assemblies, performed viral binning, and all downstream analyses. M.V.L and F.K. analyzed viral AMGs. C.M. developed software for analyses. M.V.L. conducted data validation, curation, analysis, created visualizations, and administered the project. M.V.L. and K.A. wrote the manuscript. All authors reviewed the results, edited, and approved the manuscript.

## Competing interests

The authors declare no competing interests.
