## [Transparent Peer Review file · Nature Communications]

Endemism shapes viral ecology and evolution in globally distributed hydrothermal vent ecosystems

Corresponding Author: Dr Karthik Anantharaman

Version 0:

Reviewer comments:

Reviewer #1

(Remarks to the Author)

Comment by Claire Geslin:

In this study, the authors explore the endemism that shapes viral ecology and evolution in globally distributed hydrothermal vent ecosystems. This is a very interesting topic, as the diversity and function of viruses and proviruses in deep sea hydrothermal ecosystems remain largely unexplored. This study complements and supports the few studies on the subject, including, for example, two recently works published and referenced in this manuscript (Thomas E. et al., 2021 and Cheng R. et al., 2022; references 11 and 9 respectively).

The large-scale viral metagenomics approach employed in this study is well-suited for investigating viral diversity in hydrothermal vent ecosystems. The data, methodology, and analytical techniques align with established standards in the field.

Regarding methodological considerations, analyses of viral metagenome-assembled genomes (vMAGs) revealed a predominance of bacteriophages.

While similar findings have been reported in previous studies, it is important to note that the overrepresentation of bacteriophages in vMAG analyses could be influenced by methodological biases. For instance, the VIBRANT tool used for viral identification from genomic assemblies is primarily designed for prokaryotic phages.

Furthermore, the reliance on DNA extraction and sequencing limits the scope of this study, as it excludes RNA and maybe eukaryotic viruses. That's understandable, but it needs to be specified.

Other technical points:

Have the authors cross-referenced several analyses, enabling them to support the conclusion that the identified viruses are mainly virulent, i.e. undergo a lytic cycle?

The absence of CRISPR analysis in this study raises the question of whether this approach could have offered insights into host-virus interactions.

Regarding Figure 2, the authors should provide additional context and explanation for the substantial discrepancy observed between the two analytical methods. "Nucleotide clustering indicated that no viruses were shared between hydrothermal vent plumes and deposits, whereas read mapping-based detection identified 36 such viruses ».

The discussion section should incorporate a more comprehensive context to differentiate between biologically meaningful findings and potential methodological limitations, such as sampling biases and inadequate databases for archaeal and eukaryotic viruses in these extreme marine environments.

The conclusion regarding the potential impacts of mining and anthropogenic activities on deep-sea ecosystems and the link with viral ecology should be explained in a little more detail.

Comments by Julien Lossouarn : Early Career Researcher participating in our co-review

The langwig et al. manuscript reports an interesting metagenomic study, in which hydrothermal vent prokaryotic and viral communities are investigated at a global scale. By comparing for the first time viral compositions in both vent chimney deposits and plumes, this work contributes to giving us a better overall view of viral ecology in deep-sea hydrothermal vents, which remains largely understudied. Whether the study was generally well done and the manuscript well written, it still needs to be improved by addressing the following comments.

On a semantic level :

- I would prefer you to distinguish between virulent viruses (only capable of performing a lytic cycle) and temperate viruses (capable of performing a lysogenic or a lytic cycle). Keeping in mind that this distinction virulent/temperate exclude other types of host relationships, both virulent and temperate viruses are also able to entertain pseudolysogeny while other viruses are responsible for chronic infections.

- I think a bit inappropriate to talk about viruses in the text, since most of the viral sequences are not complete and correspond to viral scaffolds or vMAGs.

Concerning the results and mat and met parts :

- You write in the results part (l.81-85) that you were able to identify 63,826 viral scaffolds from your 52 samples, from which you obtained 49,962 vMAGs after the binning step. These numbers are apparently not the same in the mat and met section, where the total number of viral scaffolds seems to be 64,220 (l.488-489).

- In my opinion, there is an ambiguity between what you call hydrothermal plume assemblies and microbial MAGs (l.468-469). By hydrothermal plume assemblies, do you mean assemblies obtained from all the 0.2 µm filtered plume viral fractions (l.432-440) ? If it is the case, what do you precisely call microbial MAGs ? Do they correspond to what you have assembled from the corresponding prokaryotic fractions ? In the same way, what you call hydrothermal deposit assemblies and microbial MAGs remains ambiguous. In this case, do you mean viruses and prokaryotes were sequenced simultaneously ? This would need to be reformulated. A little more information would be useful in the mat and met part about MAGs and how they were analysed.

- To evaluate the quality of the assembly step, it is a good idea whenever possible to calculate percentage of reads mapping onto the overall set of scaffolds, and then vMAGs.

- By the way, I wonder if it would be not interesting to challenge a bit the way in building up your viral dataset. It is indeed relatively frequent to definitively remove at least scaffolds < 2 kb, which correspond to the minimal size reported for DNA viruses (if I do not a mistake you only keep scaffolds > 3kb for many analyses). Then a clustering could be directly performed for the remaining scaffolds at the 95% ANI, corresponding to species boundaries. I would then advise the authors to use other viral prediction tools, in addition to Vibrant, such as geNomad and VirSorter2 to try to get a more exhaustive detection of what we could call viral operational taxonomic units (vOTUs).

- It would be interesting to use an additional viral database like Phrogs to annotate the viral proteins

- I advise you to be cautious about your virulent/temperate classification (l.88-92). The analysis performed on the overall dataset is not relevant given most of your viral scaffolds are 1-5 kbs. The smaller (incomplete) the scaffold, the lower the chances of detecting integrases/excisionnases/repressors. I would recommend to perform this analysis to high quality and better scaffolds only, and then including medium quality scaffolds but > 20 kb. Just to evaluate if this impacts the trend that you observe when you consider medium and better quality scaffolds. Keeping also in mind that « plasmidic » temperate viruses that do not encode integrases exist.

- I would also advise more caution about narrow vs broad host range phage conclusions given this is based solely on 14% of the viral dataset.

- l.266-270 : In my opinion, it is not all the time an easy task to clearly distinguish between AMGs and morons (by definition solely encoded by temperate viruses and which the expression provides an advantage to the lysogenized host), particularly in this case.

- Figure 1B : (l.113) yellow circles ? Virus class names are shown on the top x axis and the horizontal names above them show virus realms but Revtraviricetes is a class, not a realm. Ribozyviria are missing ?

(l. 161) family names has to be written in italic

Reviewer #2

(Remarks to the Author)

Title: Endemism shapes viral ecology and evolution in globally distributed hydrothermal vent ecosystem

Summary

In this study, Langwig et al. studied viral ecology and evolution from reconstructed viral communities in deep-sea hydrothermal environments globally in the ocean using metagenomic sequences. The authors explored the biogeographical distribution patterns of these viruses and concluded most were endemic, though a few were shared between vents. The authors evaluated inter- and intra-vent viral relatedness based on sequence similarity and read abundance. Furthermore, they explored function roles of these viruses based on protein and auxiliary metabolic genes (AMGs), as well as inferring their probable hosts.

We commend the authors for providing global context to deep-sea hydrothermal vent viruses and using state-of-the-art analytics to explore these viruses' ecological and functional patterns. Overall, the study is well-written and presents deep-sea hydrothermal-associated viruses globally. However, there are several concerns that the authors need to address, which are outlined:

Comments

- The authors claimed endemism of hydrothermal vent viruses by comparing viruses within hydrothermal environments. However, it may be confusing to state endemism without comparing the viruses with those identified in the global oceans and deep ocean environments from published work.

Line 18: What taxonomic rank are the 49,962 vOTU ~species?

Line 24: What are the metabolic functional roles of the AMGs? Briefly state this.

Line 86-87: The authors describe three states of the vMAGs, high, medium, and low confidences, what parameters/criteria was used to assign them to each of these categories. I could not find this in the MM

- How do the 49,962 viruses identified in this study compare with the Global Ocean Viromes dataset and viruses reported from deep ocean environments?

Line 88-89: The authors provide evidence of highly lytic compared to lysogenic states based on the presence of the integrase gene. Given that these findings contradict the commonly held paradigm, as the authors define in the introduction (Line 55), the authors need to provide biological justification for why they observe different findings.

->There needs to be a limitation paragraph, given this only based on evidence of a single gene.

Figure 1: Make A and B as a panel rather than standalone figures.

Line 100: The authors should include the depth information in Figure 1A.

Line 103: Could you switch the order of the realm Varidnaviria and Unknown on Figure 1B, making the unknown the last column?

-> Also unknown could be multiple viruses, so those should be in abundance figure.

Line 117-129: While the biogeographical distribution patterns are primarily based on clustering, it seems read mapping was only done on a subset of those found in multiple areas. I suggest the authors do a read mapping using their ~49k viruses and compare how this matches the geographical patterns of clustering.

- This is mainly because the genome length will influence the clustering, thus likely missing out on some viruses

Line 201: Why use a 3kb fraction rather than 5 kb or the standard 10kb for viral ecology work? How many viral hallmark genes are present within a 3kb?

- Given that viral ecology is determined at 10kb, I would recommend if the authors want to keep the read mapping at 3kb to provide benchmarking showing that larger fragment (i.e., the standard 10kb) would not have affected their conclusions.

- In addition, given that these are total metagenomes rather than enriched viral metagenomes, I would prefer larger viral contigs to reduce the risk of cellular contamination.

Figure 3: Complement this with panel B of the heatmap and clustering pattern showing protein function and biogeographical distribution patterns.

Line 252: I recommend moving this below to allow completion of the biogeography patterns/and story then discuss the AMGs story.

Line 259: Please add what DRAM-v score cut-off applied for AMG identification. Were there efforts to curate this manually?

Line: 289: A possible clear way would be to add the biogeographic information and abundance patterns within Figure 1 and then tell the story of the viruses identified and their biogeographic patterns there. Currently, where it is within the manuscript feels disjointed.

Line 303: Need some statistical correlation analysis to support this conclusion, and the results from Figure 4

Line 337: The authors need to add a limitation paragraph, given the limitations of some of the approaches they decided to use. This would provide the reader with a full context of the work and factors to keep in mind as they read the manuscript.

For example, examining viral lifestyle remains an area of active research; the current approach of using a single gene, i.e., the integrase, is highly dependent on the genome fragments, especially in the case of metagenomes. Secondly, the authors use smaller fragments > 3kb; the field standard seems to be around 10kb for viral ecology (read mapping to establish the abundance and distribution patterns).

Line 336-338: As mentioned above, a statistical analysis is needed to establish a correlation between viral and microbial abundance. Currently, what is present in Fig 4 is a qualitative view. It can remain, but adding a panel with correlations is recommended to give a robust conclusion.

Line 401-404: The authors need to provide some hypothesis or explanation for why AMGs that ideally provide niche specialization are absent within the dataset. What is the plausible explanation or mechanism that virus-host pairs are utilizing within this extreme ecosystem?

For ease of reading the supplementary tables, could the authors combine these supplementary tables into one workbook and provide the first tab as an outline of what each tab contains? In its current form, it is extremely difficult to track as one reads the manuscript.

(Remarks to the Author)

Reviewer #4

(Remarks to the Author)

Version 1:

Reviewer comments:

Reviewer #1

(Remarks to the Author)

Dear Editor and Authors,

The authors have demonstrated careful consideration of our comments.

Each point of discussion has been addressed, and the manuscript has been revised accordingly where appropriate.

In light of the incorporated new information and the implemented modifications, we believe the manuscript now meets the Nature Communications's publication criteria.

Signed : Reviewer 1 and early career researcher participating in our co-review.

Reviewer #2

(Remarks to the Author)

I thank the authors for thoroughly addressing my concerns as well as the additional analysis that they undertook. I have no further comments or issues and I am happy to see this paper published.

Reviewer #3

(Remarks to the Author)

POINT BY POINT RESPONSES TO REVIEWER COMMENTS

Dear Reviewers,

Thank you for your insightful comments on our work. We are glad the reviewers found our manuscript well-written, and our results clearly presented. We have made minor and major revisions to address the reviewers' concerns. All line numbers mentioned here refer to the manuscript with tracked changes. Author responses to reviewer comments are in blue type.

On behalf of all authors,

Sincerely,
Marguerite Langwig, *first author*
Karthik Anantharaman, *corresponding author*

REVIEWER COMMENTS

Reviewer #1 (Remarks to the Author):

Comment by Claire Geslin:

In this study, the authors explore the endemism that shapes viral ecology and evolution in globally distributed hydrothermal vent ecosystems. This is a very interesting topic, as the diversity and function of viruses and proviruses in deep sea hydrothermal ecosystems remain largely unexplored. This study complements and supports the few studies on the subject, including, for example, two recently works published and referenced in this manuscript (Thomas E. et al., 2021 and Cheng R. et al., 2022; references 11 and 9 respectively).

The large-scale viral metagenomics approach employed in this study is well-suited for investigating viral diversity in hydrothermal vent ecosystems. The data, methodology, and analytical techniques align with established standards in the field.

Regarding methodological considerations, analyses of viral metagenome-assembled genomes (vMAGs) revealed a predominance of bacteriophages.

While similar findings have been reported in previous studies, it is important to note that the overrepresentation of bacteriophages in vMAG analyses could be influenced by methodological biases. For instance, the VIBRANT tool used for viral identification from genomic assemblies is primarily designed for prokaryotic phages.

Furthermore, the reliance on DNA extraction and sequencing limits the scope of this study, as it excludes RNA and maybe eukaryotic viruses. That's understandable, but it needs to be specified.

We have added a section about methodological biases in the discussion, including a limitations paragraph as suggested by the reviewer.

Addressed at lines 508-534.

Other technical points:

Have the authors cross-referenced several analyses, enabling them to support the conclusion that the identified viruses are mainly virulent, i.e. undergo a lytic cycle?

Yes, we used multiple tools to validate and support our conclusions of the abundance of lytic viruses. In addition to VIBRANT, we ran CheckV and geNomad, which provide predictions of lytic and lysogenic states. CheckV, geNomad, and VIBRANT predicted 919, 181, and 2391 lysogenic viruses respectively. While these numbers seem to indicate a large range, they all suggest an overwhelming dominance of lytic viruses. We chose to use the VIBRANT results, which estimated the highest number of lysogenic viruses among these tools (2,391

lysogenic viruses when considering the entire dataset). VIBRANT is highly accurate in its prediction of lysogenic viruses using its “V-score metric” based approach.

The absence of CRISPR analysis in this study raises the question of whether this approach could have offered insights into host-virus interactions.

To clarify, CRISPR spacer matches were indeed used as part of our host prediction approach. We used the iPHoP tool that leverages CRISPR analysis (in addition to other approaches such as k-mer and tRNA matches) as part of the host prediction strategy. iPHoP is the most robust and comprehensive tool currently available for assessing virus-host linkages. We agree that CRISPR matches between viruses and hosts offer the highest confidence and this is factored into a weighted strategy by iPHoP’s algorithm. To make this clear in the manuscript, we have added an additional sentence explaining this in the methods under “Host Prediction.” Thank you for this comment.

Addressed at lines 747-749.

Regarding Figure 2, the authors should provide additional context and explanation for the substantial discrepancy observed between the two analytical methods. “Nucleotide clustering indicated that no viruses were shared between hydrothermal vent plumes and deposits, whereas read mapping-based detection identified 36 such viruses ».

The reasoning behind the discrepancy between these methods is stated in the second paragraph of the results section ‘Viruses are rarely shared between geographically distant hydrothermal vents or the global ocean’ as, “Some shared viruses may be missed when examining viruses identified from metagenomes, because viral sequences may only be present in the reads and not the assemblies.” Nucleotide clustering compares viruses at the genome level and hence can only be used for assembled viruses – i.e., for comparisons of viruses that are present in the samples above a certain threshold abundance. The viruses that were only identified via read mapping may be present in the reads of another sample but were never assembled. This can happen for a variety of reasons, including uneven coverage, errors in the reads, or repetitive DNA, which all make it difficult for assemblers to resolve paths through the assembly graph.

Addressed at lines 146-148.

The discussion section should incorporate a more comprehensive context to differentiate between biologically meaningful findings and potential methodological limitations, such as sampling biases and inadequate databases for archaeal and eukaryotic viruses in these extreme marine environments.

Thank you for pointing this out, we have now added a paragraph in the discussion addressing concerns related to focusing mainly on prokaryote-infecting viruses, viral lifestyle predictions, and AMG identification.

Addressed at lines 508-534.

The conclusion regarding the potential impacts of mining and anthropogenic activities on deep-sea ecosystems and the link with viral ecology should be explained in a little more detail.

To clarify, this was added at the end of the discussion: “Specifically, mining activities will disrupt natural fluid flow from hydrothermal systems, potentially reducing the availability of chemical energy for primary producing microbial chemolithoautotrophs, heterotrophs, and the larger food web. This will affect the viruses that infect these organisms.”

Addressed at lines 540-543.

Comments by Julien Lossouarn : Early Career Researcher participating in our co-review

The langwig et al. manuscript reports an interesting metagenomic study, in which hydrothermal vent prokaryotic and viral communities are investigated at a global scale. By comparing for the first time viral compositions in both

vent chimney deposits and plumes, this work contributes to giving us a better overall view of viral ecology in deep-sea hydrothermal vents, which remains largely understudied. Whether the study was generally well done and the manuscript well written, it still needs to be improved by addressing the following comments.

On a semantic level :

- I would prefer you to distinguish between virulent viruses (only capable of performing a lytic cycle) and temperate viruses (capable of performing a lysogenic or a lytic cycle). Keeping in mind that this distinction virulent/temperate exclude other types of host relationships, both virulent and temperate viruses are also able to entertain pseudolysogeny while other viruses are responsible for chronic infections.

The number of lytic and lysogenic viruses is stated in the results at line 91, “Most of the hydrothermal vent viruses were classified as lytic rather than lysogenic (47,571 lytic versus 2,391 lysogenic)” and throughout the rest of the results, when relevant. We have also added an explanation in the discussion of why this distinction can be difficult when examining incomplete viral genomes (lines 508-532).

Addressed at lines 91 and 508-532.

- I think a bit inappropriate to talk about viruses in the text, since most of the viral sequences are not complete and correspond to viral scaffolds or vMAGs.

We disagree with the reviewer here. Although many of the viruses are incomplete and identified from metagenomic data, this is standard in the field to refer to the putative viruses as “virus” (see Tian et al., 2024 and Roux et al., 2021 for examples). Numerous studies and computational tools have been benchmarked demonstrating high confidence in viral scaffolds originating from viruses.

Tian, F., Wainaina, J. M., Howard-Varona, C., Domínguez-Huerta, G., Bolduc, B., Gazitúa, M. C., ... & Sullivan, M. B. (2024). Prokaryotic-virus-encoded auxiliary metabolic genes throughout the global oceans. *Microbiome*, 12(1), 159.

Roux, S., Matthijnssens, J., & Dutilh, B. E. (2021). Metagenomics in virology. *Encyclopedia of Virology*, 133.

Concerning the results and mat and met parts :

- You write in the results part (l.81-85) that you were able to identify 63,826 viral scaffolds from your 52 samples, from which you obtained 49,962 vMAGs after the binning step. These numbers are apparently not the same in the mat and met section, where the total number of viral scaffolds seems to be 64,220 (l.488-489).

Thank you for catching this, this was a mistake and has been corrected with the correct number of scaffolds.

Addressed at line 621.

- In my opinion, there is an ambiguity between what you call hydrothermal plume assemblies and microbial MAGs (l.468-469). By hydrothermal plume assemblies, do you mean assemblies obtained from all the 0.2 µm filtered plume viral fractions (l.432-440) ? If it is the case, what do you precisely call microbial MAGs ? Do they correspond to what you have assembled from the corresponding prokaryotic fractions ? In the same way, what you call hydrothermal deposit assemblies and microbial MAGs remains ambiguous. In this case, do you mean viruses and prokaryotes were sequenced simultaneously ? This would need to be reformulated. A little more information would be useful in the mat and met part about MAGs and how they were analysed.

The plume assemblies are largely obtained from 0.2 µm filters, as well as 0.8 µm filters. The DNA was extracted from these filters and sequenced to produce metagenomes. Both microbial MAGs and viruses are then identified and reconstructed from these metagenomes. For hydrothermal deposits, no filter is necessary as these samples are taken from the chimney deposits directly. DNA is extracted from deposit material and sequenced to produce metagenomes. Both microbial MAGs and viruses (including vMAGs) are identified and reconstructed from these metagenomes.

Assemblies are the assembled contigs of metagenomic reads. MAGs are the metagenome-assembled genomes that are reconstructed from the assembly after read mapping and binning. MAGs can be microbial or viral (vMAGs). Microbial MAGs and vMAGs can both be obtained from the same assembly.

- To evaluate the quality of the assembly step, it is a good idea whenever possible to calculate the percentage of reads mapping onto the overall set of scaffolds, and then vMAGs.

The assemblies are all previously published and have been assessed for quality in those publications (Zhou et al., 2022 and Zhou et al., 2023). Here, because we are reusing these data to examine viruses, we completed the read mapping to viral genomes and provide the data in the supplementary tables.

Zhou, Z., St John, E., Anantharaman, K. & Reysenbach, A.-L. Global patterns of diversity and metabolism of microbial communities in deep-sea hydrothermal vent deposits. *Microbiome* 10, 241 (2022).

Zhou, Z. et al. Sulfur cycling connects microbiomes and biogeochemistry in deep-sea hydrothermal plumes. *ISME J.* (2023) doi:10.1038/s41396-023-01421-0.

- By the way, I wonder if it would be not interesting to challenge a bit the way in building up your viral dataset. It is indeed relatively frequent to definitively remove at least scaffolds < 2 kb, which correspond to the minimal size reported for DNA viruses (if I do not a mistake you only keep scaffolds > 3kb for many analyses). Then a clustering could be directly performed for the remaining scaffolds at the 95% ANI, corresponding to species boundaries. I would then advise the authors to use other viral prediction tools, in addition to Vibrant, such as geNomad and VirSorter2 to try to get a more exhaustive detection of what we could call viral operational taxonomic units (vOTUs).

We agree with the reviewer that this is an exciting area of viromics research and will be interesting to incorporate in future studies. However, incorporating this is currently beyond the scope our study. While using multiple tools would likely yield a few more viruses that would be added to our dataset, most of the viruses identified with multiple tools are redundant. The central conclusion of the paper, that viruses are largely endemic in hydrothermal vents, is unlikely to change with the addition of some viruses, given the data and statistics are apparent from thousands of these genomes. It is relatively common to still use one viral prediction tool (as seen in these recent examples VirSorter2: Simon et al., 2023 and Tian et al., 2024) and the benefits of using multiple virus prediction tools and how they should be combined is still an active area of research.

Lopez-Simon, J., Vila-Nistal, M., Rosenova, A., De Corte, D., Baltar, F., & Martinez-Garcia, M. (2023). Viruses under the Antarctic Ice Shelf are active and potentially involved in global nutrient cycles. *Nature Communications*, 14(1), 8295.

Tian, F., Wainaina, J. M., Howard-Varona, C., Domínguez-Huerta, G., Bolduc, B., Gazitúa, M. C., ... & Sullivan, M. B. (2024). Prokaryotic-virus-encoded auxiliary metabolic genes throughout the global oceans. *Microbiome*, 12(1), 159.

- It would be interesting to use an additional viral database like Phrogs to annotate the viral proteins

We have now added this into Supplementary Table 11 as requested. These annotations are incorporated into our results and support our conclusion that most viral proteins are predicted to be hypothetical or of unknown function.

Addressed at lines 239-241.

- I advise you to be cautious about your virulent/temperate classification (1.88-92). The analysis performed on the overall dataset is not relevant given most of your viral scaffolds are 1-5 kbs. The smaller (incomplete) the scaffold, the lower the chances of detecting integrases/excisionnases/repressors. I would recommend to perform this analysis to high quality and better scaffolds only, and then including medium quality scaffolds but > 20 kb. Just to evaluate if this impacts the trend that you observe when you consider medium and better quality scaffolds. Keeping also in mind that « plasmidic » temperate viruses that do not encode integrases exist.

We agree and have added information in the discussion describing this limitation. We find the pattern holds when examining medium-quality and above viruses, and when examining only complete viruses. This information has been added to the first paragraph of the results, where the entire sentence now reads, “Most of the hydrothermal vent viruses were classified as lytic rather than lysogenic (47,571 lytic versus 2,391 lysogenic) and this remains true when only examining viruses of medium-quality or better (1,505 lytic versus 328 lysogenic), as well as only complete viruses (109 lytic versus 4 lysogenic).”

Addressed at lines 92-93.

- I would also advise more caution about narrow vs broad host range phage conclusions given this is based solely on 14% of the viral dataset.

We added “and for these viruses,” at the beginning of the results section ‘Viral biogeography is closely tied to the geographic distribution and abundance of their hosts’ to reinforce that this is just for the 14% with a predicted host.

Addressed at line 288.

- l.266-270 : In my opinion, it is not all the time an easy task to clearly distinguish between AMGs and morons (by definition solely encoded by temperate viruses and which the expression provides an advantage to the lysogenized host), particularly in this case.

Our goal was solely to highlight genes coopted by viruses from their hosts and we only refer to AMGs in this manuscript. We do not differentiate between morons and AMGs. For all practical purposes, morons are a subset of AMGs and current standards in viromics do not differentiate between these two designations. We agree with the reviewer that differentiating between AMGs and morons as defined by the reviewer is difficult and may need experimental validation, which is currently beyond the scope of our study.

- Figure 1B : (l.113) yellow circles ? Virus class names are shown on the top x axis and the horizontal names above them show virus realms but Revtraviricetes is a class, not a realm. Ribozyviria are missing ?

Thank you, good catch, this was from a previous color scheme for that figure and has been corrected to teal (line 118).

And thank you for the second catch, something happened with the name here and it has been corrected - the Revtraviricetes class is within the Riboviria realm, and this name is now with the other realm names (line 103, Figure 1B).

Addressed at line 103 and 118.

(l. 161) family names has to be written in italic

Thank you for catching this, we have italicized and checked the rest of the manuscript for this formatting (line 179-180).

Addressed at lines 179-180.

Reviewer #2 (Remarks to the Author):

Title: Endemism shapes viral ecology and evolution in globally distributed hydrothermal vent ecosystem

Summary

In this study, Langwig et al. studied viral ecology and evolution from reconstructed viral communities in deep-sea hydrothermal environments globally in the ocean using metagenomic sequences. The authors explored the biogeographical distribution patterns of these viruses and concluded most were endemic, though a few were

shared between vents. The authors evaluated inter- and intra-vent viral relatedness based on sequence similarity and read abundance. Furthermore, they explored function roles of these viruses based on protein and auxiliary metabolic genes (AMGs), as well as inferring their probable hosts.

We commend the authors for providing global context to deep-sea hydrothermal vent viruses and using state-of-the-art analytics to explore these viruses' ecological and functional patterns. Overall, the study is well-written and presents deep-sea hydrothermal-associated viruses globally. However, there are several concerns that the authors need to address, which are outlined:

Comments

- The authors claimed endemism of hydrothermal vent viruses by comparing viruses within hydrothermal environments. However, it may be confusing to state endemism without comparing the viruses with those identified in the global oceans and deep ocean environments from published work.

We ran the same nucleotide clustering analysis on the vent viruses with the Global Ocean Virome 2.0 dataset and identified only 36 clusters to be shared between our hydrothermal vent samples and GOV. We have now added a description of these results in this section now entitled 'Viruses are rarely shared between geographically distant hydrothermal vents or the global ocean' (lines 155-166). This information is also provided in Supplementary Table 9.

Addressed at lines 155-166.

Line 18: What taxonomic rank are the 49,962 vOTU ~species?

The taxonomic predictions for the viral genomes were generated using geNomad and are available in Supplementary Table 2.

Line 24: What are the metabolic functional roles of the AMGs? Briefly state this.

This has been added in the abstract at line 25. To clearly connect this statement to the results, we also added a brief explanation of these genes in the results, at the end of the section 'Hydrothermal vents share viral protein families dominated by proteins of unknown function' (lines 258-262).

Addressed at lines 25 and 258-262.

Line 86-87: The authors describe three states of the vMAGs, high, medium, and low confidences, what parameters/criteria was used to assign them to each of these categories. I could not find this in the MM

This was predicted using CheckV and is stated in the methods section 'Virus taxonomy, marker genes, and quality'. We have added clarification to this sentence in the methods explicitly stating that the low-, medium-, and high-quality as well as complete predictions come from this tool (lines 651-652).

Addressed at lines 651-652.

- How do the 49,962 viruses identified in this study compare with the Global Ocean Viromes dataset and viruses reported from deep ocean environments?

As stated above, we ran the same nucleotide clustering analysis on the vent viruses with the Global Ocean Virome 2.0 dataset and identified only 36 clusters in total where vent viruses have $\geq 70\%$ nucleotide identity to viruses from GOV (lines 155-166).

Addressed at lines 155-166.

Line 88-89: The authors provide evidence of highly lytic compared to lysogenic states based on the presence of the integrase gene. Given that these findings contradict the commonly held paradigm, as the authors define in the introduction (Line 55), the authors need to provide biological justification for why they observe different findings.

->There needs to be a limitation paragraph, given this only based on evidence of a single gene.

We agree and have added a limitations paragraph (lines 508-534) in the discussion stating the issues in predicting lifestyle introduced by incomplete viral genomes. We also note in the results that this pattern was observed using medium-quality and above, as well as complete viral genomes (though the number of complete viral genomes in our study is low).

Addressed at lines 508-534.

Figure 1: Make A and B as a panel rather than standalone figures.

This is now adjusted in Figure 1 (line 103).

Addressed at line 103.

Line 100: The authors should include the depth information in Figure 1A.

Thank you for this suggestion, depth information is now included in the map (line 103).

Addressed at line 103.

Line 103: Could you switch the order of the realm Varidnaviria and Unknown on Figure 1B, making the unknown the last column?

Yes, thank you, this has been changed and the figure replaced (line 103).

Addressed at line 103.

-> Also unknown could be multiple viruses, so those should be in abundance figure.

The abundance of all unknown viruses is in the last column as suggested (line 103).

Addressed at line 103.

Line 117-129: While the biogeographical distribution patterns are primarily based on clustering, it seems read mapping was only done on a subset of those found in multiple areas. I suggest the authors do a read mapping using their ~49k viruses and compare how this matches the geographical patterns of clustering.

- This is mainly because the genome length will influence the clustering, thus likely missing out on some viruses

To be clear, for both ANI-based clustering and read mapping-based clustering, the analyses were completed on $\geq 3\text{kb}$ viruses, and thus both were completed on the same set of viruses.

We did not complete these analyses on the full set because viruses shorter than this length drastically reduced noise and when manually examined, showed meaningful overlap in their annotations. To clarify this, we have added " $\geq 3\text{kb}$ " in the results section (line 127), 'Viruses are rarely shared between geographically distant hydrothermal vents or the global ocean.' This is also stated in the figure caption of Figure 2 (line 213) and was corrected in the methods, line 732.

Addressed at lines 127, 213, and 732.

Line 201: Why use a 3kb fraction rather than 5 kb or the standard 10kb for viral ecology work? How many viral hallmark genes are present within a 3kb?

In our opinion, 10kb is an extremely strict cut off - while this ensures you have less to worry about in terms of manually checking viruses it will certainly exclude real viruses and viral fragments, considering the smallest bacteriophages are 3kb (Rumnieks and Tars, 2012). In support of this, we observed several 3kb

Cressdnaviricota and Microviridae in our dataset that are predicted to be complete viral genomes. Since we planned to use a genome clustering comparison method to describe viral biogeography, we wanted to be sure to capture as much data as possible - if we had only included 10kb viruses we would have missed additional related viruses that are fragmented or have a smaller genome size. Considering that we include these viruses and still identify so few shared viruses across geographic distances is telling.

In this dataset, 3kb viruses have 0-8 viral hallmark genes, according to geNomad. This was why we were very careful in reporting an abundance of metadata with the viruses. The supplementary data includes all the information we have about the viruses that were found to overlap (gene annotations, estimates of completeness, genome size, quality, etc.), and the annotations of the regions of overlap. For example, in cluster 12 that includes 3-4kb viruses, the regions of overlap include base plate wedge proteins. While we agree that the inferences we can make about these viruses outside of identifying these regions of overlap is limited, we think it is important to include them to ensure we are capturing as many viruses as possible, and their exclusion would only reinforce the central theme of the paper, that vent viruses are largely endemic.

Rumnieks, J., & Tars, K. (2012). Diversity of pili-specific bacteriophages: genome sequence of IncM plasmid-dependent RNA phage M. *BMC microbiology*, 12, 1-8.

- Given that viral ecology is determined at 10kb, I would recommend if the authors want to keep the read mapping at 3kb to provide benchmarking showing that larger fragment (i.e., the standard 10kb) would not have affected their conclusions.
- In addition, given that these are total metagenomes rather than enriched viral metagenomes, I would prefer larger viral contigs to reduce the risk of cellular contamination.

This information can be seen in the supplementary data, because 10 kb and above viruses were included in all analyses - if you were to place the cut off at 10kb, there would only be 15 ANI-based clusters with viruses from geographically distinct vents, as opposed to 65. Thus, the current conclusions are further reinforced, that endemism dominates.

Concerns about viral contig size are addressed in your comment above.

Figure 3: Complement this with panel B of the heatmap and clustering pattern showing protein function and biogeographical distribution patterns.

Thank you for this suggestion, we have now added panel B showing the annotations of the proteins from part A (line 267). There are thousands of proteins within the clusters shown in panel A, thus the functional categories shown in B are broad because it would be impossible to display all the individual protein names. We also normalized the protein counts (number of times a particular protein occurs at a site) by the total number of annotated proteins at a site to better show the patterns of protein distributions between sites.

Addressed at line 267.

Line 252: I recommend moving this below to allow completion of the biogeography patterns/and story then discuss the AMGs story.

Thank you for this suggestion, to improve flow, the AMGs are now last in the results and the patterns with host abundance and hydrothermal geology are now before it (line 282).

Addressed at line 282.

Line 259: Please add what DRAM-v score cut-off applied for AMG identification. Were there efforts to curate this manually?

To address this, we have clarified language in the main text (lines 353-355) and added more detailed information in the methods (lines 719-730).

Addressed at lines 353-355 and 719-730.

Line: 289: A possible clear way would be to add the biogeographic information and abundance patterns within Figure 1 and then tell the story of the viruses identified and their biogeographic patterns there. Currently, where it is within the manuscript feels disjointed.

Per your above suggestion, the AMG section is now last in the results, improving the flow of this section (line 347).

Addressed at line 347.

Line 303: Need some statistical correlation analysis to support this conclusion, and the results from Figure 4.

Thank you for your feedback, this information was added to the results (lines 321-327) and supplementary data (Supplementary Table 15). Proportionality was calculated using the propr package in R and the methods describing this are detailed in the methods section 'Host prediction' (lines 757-763).

Quinn, T. P., Richardson, M. F., Lovell, D., & Crowley, T. M. (2017). propr: an R-package for identifying proportionally abundant features using compositional data analysis. *Scientific reports*, 7(1), 1-9.

Addressed at lines 321-327 and 757-763.

Line 337: The authors need to add a limitation paragraph, given the limitations of some of the approaches they decided to use. This would provide the reader with a full context of the work and factors to keep in mind as they read the manuscript.

For example, examining viral lifestyle remains an area of active research; the current approach of using a single gene, i.e., the integrase, is highly dependent on the genome fragments, especially in the case of metagenomes. Secondly, the authors use smaller fragments > 3kb; the field standard seems to be around 10kb for viral ecology (read mapping to establish the abundance and distribution patterns).

We agree, and this was mentioned by another reviewer, thank you for bringing this up. We have added this paragraph in the discussion, where we highlight concerns related to focusing on mainly prokaryote-infecting viruses, viral lifestyle predictions, and AMG identification (lines 507-531).

Addressed at lines 507-531.

Line 336-338: As mentioned above, a statistical analysis is needed to establish a correlation between viral and microbial abundance. Currently, what is present in Fig 4 is a qualitative view. It can remain, but adding a panel with correlations is recommended to give a robust conclusion.

As stated above, proportionality was calculated to establish that this correlation is supported.

Addressed at lines 321-327 and 757-763.

Line 401-404: The authors need to provide some hypothesis or explanation for why AMGs that ideally provide niche specialization are absent within the dataset. What is the plausible explanation or mechanism that virus-host pairs are utilizing within this extreme ecosystem?

We have now added a more comprehensive explanation of the limitations of AMG predictions in the discussion that addresses this comment.

Addressed at lines 507-531.

For ease of reading the supplementary tables, could the authors combine these supplementary tables into one workbook and provide the first tab as an outline of what each tab contains? In its current form, it is extremely difficult to track as one reads the manuscript.

Thank you for this suggestion, this helps improve readability a lot. All the tables are now in one excel file under multiple sheets with a table of contents at the beginning. We have updated the supplementary table numbers throughout the main text.

Reviewer #3 (Remarks to the Author):

We thank this reviewer for their support of early career researchers.

Reviewer #4 (Remarks to the Author):

We thank this reviewer for their support of early career researchers.